# Detection of tar brown carbon with the single particle soot photometer (SP2)

Joel C. Corbin [1] and Martin Gysel-Beer [2]

[1]Metrology Research Centre, 1200 Montreal Road, National Research Council Canada, Ottawa, ON K1A 0R6, Canada
[2]Laboratory of Atmospheric Chemistry, Paul Scherrer Institute, 5232 Villigen PSI, Switzerland

**Correspondence:** Joel Corbin (Joel.Corbin@nrc-cnrc.gc.ca)

**Abstract.**

We investigate the possibility that the refractory, infrared-light-absorbing carbon particulate material known as "tarballs" or tar brown carbon (tar brC) generates a unique signal in the scattering and incandescent detectors of the single particle soot photometer (SP2). As recent studies have defined tar brC in different ways, we begin by reviewing the literature and proposing a material-based definition of tar. We then show that tar brC results in unique SP2 signals due to a combination of complete or partial evaporation, with no or very little incandescence. Only a subset of tar brC particles exhibited detectable incandescence (70% by number); for these particles the ratio of incandescence to light scattering was much lower than that of soot BC. At the time of incandescence the ratio of light scattering to incandescence from these particles was up to twofold greater than from soot black carbon (BC). In our sample, where the mass of tar was threefold greater than the mass of soot, this led to a bias of $< 5\%$ in SP2-measured soot mass, which is negligible relative to calibration uncertainties. The enhanced light scattering of tar is interpreted as due to its being more amorphous and less graphitic than soot BC. The fraction of the tar particle which does incandesce was likely formed by thermal annealing during laser heating.

These results indicate that laser-induced incandescence, as implemented in the SP2, is the only BC measurement technique which can quantify soot BC concentrations separately from tar, while also potentially providing real-time evidence for the presence of tar. In contrast, BC measurement techniques based on thermal–optical ("EC") and absorption ("eBC") measurements cannot provide such distinctions. The optical properties of our tar particles indicate a material similarity to the tar particles previously reported in the literature. However, more- and less-graphitized tar samples have also been reported, which may show stronger and weaker SP2 responses, respectively.

## 1 Introduction

Atmospheric light-absorbing carbon (LAC) in particulate matter (PM) plays a substantial role in the radiative balance of the earth both directly and by influencing cloud properties (Boucher et al., 2013). While soot black carbon (soot BC) is the best-recognized form of LAC (Bond et al., 2013), increasing attention has recently been paid to the so-called "brown carbon" (Kirchstetter et al., 2004; Laskin et al., 2015) and "tarballs" (Pósfai et al., 2004; Hand et al., 2005; Niemi et al., 2006; Semeniuk et al., 2006; Tivanski et al., 2007; Alexander et al., 2008; Vernooij et al., 2009; Chakrabarty et al., 2010; Adachi and Buseck, 2011;

China et al., 2013; Zhu et al., 2013; Tóth et al., 2014; Hoffer et al., 2016a, b; Sedlacek III et al., 2018; Corbin et al., 2019) which possess substantially different physical properties than BC. The term "brown carbon" is canonically used to refer to the collection of substantially light-absorbing organic molecules found in PM, while the term "tarballs" refers to the insoluble amorphous-carbon spheres which may be produced by the pyrolysis of high-molecular-weight fuels such as biomass (Tóth et al., 2014) or heavy fuel oil (Corbin et al., 2019). Here we will refer to these two sub-types of brown carbon as "soluble brown

carbon" (soluble brC) and "tar brC" following Corbin et al. (2019). Both forms of brC may comprise a large fraction of the light absorption of atmospherically-relevant aerosols such as wildfire smoke (Lack et al., 2012; China et al., 2013; Liu et al., 2015) and marine-engine exhaust (Corbin et al., 2018b, 2019).

The distinction between soluble brC, tar, and soot BC is important as it may result in unique environmental fates and impacts of these distinct types of LAC, due to their unique chemical and morphological properties (Corbin et al., 2019). Similarly,

"BC" instruments designed to measure soot BC based on one of its defining properties – insolubility in water and organic solvents; refractoriness up to $\sim 4000$ K; a structure consisting primarily of $sp^2$-bonded graphene-like carbon; and a morphology of aggregated monomers of diameter 10–80 nm (Bond et al., 2013; Petzold et al., 2013) – may be cross-sensitive to soluble brC or tar. Cross-sensitivities to soluble brC are generally limited to the BC instruments measuring light absorption at shorter visible wavelengths of 300–500 nm, which normally report their measurements as equivalent BC (eBC, Petzold et al. 2013).

Cross-sensitivities to tar brC are more problematic, and include both eBC measured at visible and near-infrared wavelengths of 300-1000 nm, as well as and thermal–optically defined elemental carbon (EC) (Corbin et al., 2019).

The fact that tar brC may absorb substantially at wavelengths of 1000 nm implies a further potential cross-sensitivity of instruments such as the single-particle soot photometer (SP2), which relies upon a continuous-wave 1064 nm Nd:YAG laser to heat particles to incandescence. The SP2 is normally used to report single-particle rBC mass concentrations by calibrating

incandescence signals with reference to BC particles of known mass and composition. Therefore, any cross-sensitivities of SP2 rBC to tar brC would require that tar be refractory enough to reach the $>3000$ K attained by soot BC.

However, an SP2 cross sensitivity to tar brC may also occur indirectly, if a tar brC sample is capable of undergoing complete or partial annealing to rBC during heating by the SP2 laser. This would result in an incandescence signal from rBC which did not exist prior to the measurement. A recent study by Sedlacek et al. (2018) suggested that this effect may lead to apparent rBC

signals of up to 9% of the total particle mass, for laboratory-generated tar brC. Their tar brC measurements could not distinguish whether this signal was due to the formation of rBC during tar-brC formation versus in-SP2 annealing. Sedlacek et al. (2018) also performed experiments using nigrosin (a polyaniline-based dye) to demonstrate that laser-induced annealing may contribute 45% of the incandescence signal expected for pre-existing rBC. This percentage decreased with increasing laser power, to 25%, demonstrating that evaporation may occur more rapidly than annealing under appropriate conditions (including high laser power

density and low degree of graphitization in the starting material). It is interesting to note that Moteki and Kondo (2008) also measured nigrosin in an SP2 but observed zero laser-induced annealing. This may be due to the use of a lower laser intensity by Moteki and Kondo (2008), which was shown by Sedlacek et al. (2018) to potentially result in negligible incandescence.

Corbin et al. (2019) reported that the apparent rBC mass of tar particles (that is, the magnitude of any cross-sensitivities) produced by a marine engine operated on heavy fuel oil was negligible. Their inference was based on the fact that thermal–optical

EC (IMPROVE-A protocol) remained high while rBC signals fell to zero, under conditions where the aerosol AAE (the negative slope of a log-log plot of absorption against wavelength) was $\sim 2$, corresponding to low engine loads. They also found a negligible response of the Soot-Particle Aerosol Mass Spectrometer (SP-AMS; Onasch et al. 2011), which also relies on the absorption of a 1064 nm laser, to these tar particles. Note that any comparison of the SP2 and SP-AMS sensitivities should be made with caution, as particles experience lower pressure, shorter beam exposure times, and different laser-power densities in the latter instrument (laser power densities are unmeasured and may vary between instruments). The lack of SP2 cross-sensitivity to tar brC in Corbin et al. (2019) is likely related to the fact that tar brC is less refractory than soot BC, and therefore not capable of reaching incandescent temperatures prior to vapourization. Another potential factor, that the mass-specific absorption efficiency (MAE or "MAC") of the tar brC was 23-fold smaller than that of soot BC at 950 nm wavelength, (although it was only 5-fold smaller at 660 nm and 2-fold smaller at 370 nm). The MAE is fundamentally related to refractoriness, because both MAE and refractoriness will increase with increasing degree of carbonization of the tar (Corbin et al., 2019), as will be discussed further later.

A negligible incandescence signal in the SP2 does not mean that the SP2 is incapable of detecting tar brC, because the SP2 does not only measure time-resolved incandescence signals but also time-resolved scattering signals. Previous work has exploited the time-resolved SP2 scattering signals, particularly relative to the time of incandescence, to determine apparent rBC coating thicknesses (Gao et al., 2007; Laborde et al., 2012a) and to differentiate core-shell rBC particles from "attached" rBC particles (BC particles coagulated with, but not engulfed by, non-BC particles, Moteki et al. 2014).

From this basis, the present manuscript explores the possibility that a detailed analysis of the time-resolved signals allows the detection of tar brC particles by SP2 in terms of their (predicted) anomalous scattering signals. We also seek to quantify the potential interference of tar on SP2 incandescence. We use data from Corbin et al. (2019) for our analysis, contrasting a tar-containing sample with a tar-free sample from the same engine. The manuscript is structured as follows. In Section 2 we review previous work on the properties and definition of tar brC. In Section 3, we give a technical discussion of the SP2, present the test data set used below, and describe the analysis techniques used in this study. In Section 4, we present the results of this study.

## 2    Definition and properties of tar brC

### 2.1    Review of "tarball" properties

The particles referred to as "tarballs" or "tar brC" in the literature generally have consistent properties. Physically, tar brC exists as spheres of solid or extremely viscous carbonaceous material. The spheres may exist as isolated particles (Pósfai et al., 2004) or as aggregates (Hand et al., 2005; Girotto et al., 2018) and are of diameters 100–300 nm, approximately one order of magnitude larger than the corresponding monomer diameter in soot aggregates, 10–80 nm (Corbin et al., 2019; Adachi et al., 2019). Whereas the organic molecules (so-called soluble brC) typically addressed by brC studies (Laskin et al., 2015; Moschos et al., 2018) are generally soluble due to their small molecular sizes, tar brC may be insoluble in all solvents (Corbin et al., 2019). The small molecular sizes of soluble brC correspond to high volatilities, whereas tar brC is of extremely low

volatility, vapourizing at about 1000 K (Corbin et al., 2019). The low volatility of tar brC leads to its stability within an electron microscope (Pósfai et al., 2003). All of these properties are due to the amorphous-carbon-like molecular composition of tar brC, which reflects its formation via polymerization and carbonization reactions, as discussed further below. As polymerization and carbonization are continuous processes, these properties are therefore also continuous, and materials may be observed in the atmosphere with properties intermediate between the tarball properties specified above and the traditionally recognized form of brC, soluble brC (small light-absorbing molecules).

In terms of its elemental composition, tar brC consists primarily of carbon, but also contains hydrogen and oxygen (Tóth et al., 2018). Tar brC emitted from biomass burning may contain impurities of K, Cl, Si, and S (Pósfai et al., 2004; Adachi et al., 2019). Atmospheric tar brC spheres have also been reported which contained impurities of S and Si, but not K (Alexander et al., 2008; Zhu et al., 2013), which has been proposed as indicating an origin of residual-fuel (heavy-fuel-oil) combustion (Corbin et al., 2019).

In terms of their optical properties, tarballs may be considered a subset of brC, as their imaginary refractive index decreases with increasing wavelength (Alexander et al., 2008; Corbin et al., 2019), which results in a brown appearance at appropriate concentrations (Liu et al., 2016). However, light absorption by tar remains significant even in the near-infrared (about 900 nm wavelength) (Alexander et al., 2008; Hoffer et al., 2016a; Corbin et al., 2019). This light absorption has been described by the Tauc band-gap model (Corbin et al., 2019), which is also applicable to soluble brC (Sun et al., 2007), and which predicts a slow tailing off of absorption with increasing wavelength. Retrieved complex refractive indices for tar brC in the wavelength range 400–630 nm span a wide range, with real parts in the range of 1.6–1.9 and imaginary parts in the range 0.02–0.2 (Hand et al., 2005; Alexander et al., 2008; Chakrabarty et al., 2010). The lower values in this range are similar to those of the soluble brC emitted by biomass combustion (Kumar et al., 2018; Li et al., 2019). The higher values in this range correspond to absorption efficiencies comparable to those of soot BC (Alexander et al., 2008; Corbin et al., 2019). The broad range reflects a broad range of molecular structures, where larger, more-conjugated structures would correspond to more-efficient absorption.

In terms of its molecular structure, tar brC has been described as amorphous carbon, in reference to its disordered, non-crystalline structure and mixture of $sp^3$ to $sp^2$ bonds (Corbin et al., 2019), and as a form of polymerized carbon, since it contains large, insoluble molecules formed from smaller ones (Pósfai et al., 2004). We prefer the former term, amorphous, because the polymerization definition does not account for the possibility that tar brC is not only polymerized, but also carbonized (Tóth et al., 2018; Corbin et al., 2019). (Here we follow the IUPAC definition of carbonization as the heat-induced formation of a solid with increased carbon content, due to the elimination of other elements; Nič et al. 2009.) Amorphous carbon materials are those which are on average disordered and non-crystalline, yet which may possess smaller regions that are more ordered locally (Burket et al., 2008). Tar brC may also be described by the less-specific term macromolecule (Oberlin, 1984).

In the above discussion, we have referred to carbonization but not graphitization. Carbonization must precede graphitization, and is associated with an increase in aromaticity of the starting material (Oberlin, 1984). Increased aromaticity would correspond to an increase in viscosity (Reid et al., 2018), an increase in light-absorption efficiency, and a decrease in the AAE (Hoffer et al., 2016a; Corbin et al., 2019). However, though carbonization precedes graphitization, it is not equivalent; graphitization does not always occur to the same extent. Not all carbon materials are capable of converting to the thermodynamically favourable

allotrope of graphitic carbon upon heat treatment (that is, they are non-graphitizing; Nič et al., 2009), due to the presence of cross-linkages within the material, which prevent annealing by the alignment and stacking of aromatic moieties (Rouzaud and Oberlin, 1989; Burket et al., 2008; Oberlin et al., 2006). We note here that, according to IUPAC, the term graphitization does not strictly apply to nanoparticles like soot and tarballs, which are too small to possess planar graphene sheets.

The degree of carbonization and graphitization of atmospheric tar brC particles will depend on the exact nature of the precursor materials. For residual-fuel precursors, not only the amount but also the chemical state of sulfur will influence the result (Oberlin, 1984). The maximum temperature and heating profile experienced by the precursors will also have an effect (Hoffer et al., 2016a), and a continuous variety of properties is expected (Oberlin, 1984). Regardless, in all cases, the carbon in tar brC will be much less graphitized than the carbon in soot BC, according to its higher $sp^3/sp^2$ bonding ratio (Tóth et al., 2018; Corbin et al., 2019). Soot BC formation is not the endpoint of tar brC formation, due to limitations in the maximum possible degree of graphitization in tar brC and due to the fact that soot BC particles always exist as aggregates of spherules (Pósfai et al., 2004; Corbin et al., 2019). Therefore, while soot BC and tar brC may be described as existing on a continuum from the perspective of their light absorption properties (Saleh et al., 2018), they do *not* exist on a continuum of morphologies and should be considered as distinct materials.

The formation of tar brC has been proposed to proceed through the thermally- or chemically-induced polymerization and carbonization of high-molecular-weight organic fuels, including biomass (Pósfai et al., 2004; Tóth et al., 2014; Hoffer et al., 2016a; Sedlacek III et al., 2018) and crude-oil residual fuel (Corbin et al., 2019). The pyrolysis hypothesis has been directly demonstrated by laboratory studies for both biomass tar (Tóth et al., 2014) and residual-fuel tar (Jiang et al., 2019). In this context, we have considered these two fuels as chemically related, since the crude oil from which residual fuels originate is, fundamentally, thermally processed biomass. This chemical relationship is supported by the fundamental studies of Oberlin (1984). While it is obvious that molecular differences will exist between minimally processed biomass-burning emissions and residual-fuel emissions, the material properties of tar brC produced from either fuel appear to be similar, according to studies which have comprehensively characterized tar brC from either residual-fuel (Corbin et al., 2019) or biomass (Adler et al., 2019) combustion. (Note that although Adler et al. 2019 did not refer to their studied particles as tarballs, they characterized macromolecular, low-volatility, highly light-absorbing, spherical particles that were stable under an electron beam and therefore possessed all of the properties of tar brC without exception.)

The hypothesis that tar brC forms via atmospheric-processing is based on less-direct evidence from two aircraft-based studies. In the first study, Pósfai et al. (2003, 2004) used electron microscopy to observe tarball number fractions increasing from a negligible amount to 85% in a smouldering fire over Mozambique after about 1h of atmospheric processing. They argued that polymerization reactions similar to those observed during laboratory cloud-processing experiments were the most likely cause of this observation. It must also be mentioned that Pósfai et al. (2003) also observed a tarball number fraction of 18% for a separate unprocessed plume over South Africa, suggesting that atmospheric aging is not the only mechanism of tarball formation. In a separate study, Sedlacek III et al. (2018) also concluded that tarball number and mass fractions increased after about 2h of aging. They quantified tar ball number fractions by electron microscopy, and tarball mass fractions under the assumption that their tarballs flash-vapourized at 873 K in an aerosol mass spectrometer (which may not be true for the more-processed tarballs which

are stable in electron microscopes, as shown for residual-fuel tar particles; Corbin et al. 2019). The chemical mechanism by which photochemical aging may produce tarballs material is unclear, and should be investigated in future work.

Finally, we emphasize that many of the properties of tar described above reflect ranges of continuous properties. The processes by which smaller organic molecules are transformed into amorphous-carbon tarballs depend on the amount of aging for chemically-formed tarballs (Sedlacek III et al., 2018) and on the heat treatment temperature for thermally-formed tarballs (Hoffer et al., 2016a). These reactions result in larger, less-soluble, and more-strongly absorbing materials, yet there is no clear dividing line between the starting materials and the tarballs that result. The properties of tar are generally inter-related, and

will evolve co-dependently since they reflect the molecular transformations that define tar formation. Presently, the range most representative for atmospheric particles is poorly constrained. Laboratory studies on tarballs may generate particles under a wide range of conditions [e.g., smouldering combustion (Chakrabarty et al., 2010) versus dry distillation (Tóth et al., 2014; Li et al., 2019)] as well as by using a variety of approaches to aerosol generation, such as the presence (Tóth et al., 2014; Hoffer et al., 2006) or absence (Li et al., 2019) of a heat-shock procedure which appears to induce tarball carbonization (that is, increases

light absorption and lowers AAE; Hoffer et al. 2016a). These issues must be better constrained if their implementations in global models are to be reliable.

In spite of the wide range of properties which tarball-like materials may possess, their typical properties (strong light absorption, refractivity, and solid-like phase) warrant a distinction between tar brC and soluble brC (small light-absorbing molecules). This distinction is important for two reasons. First, *tar brC would* not *be detected by the solvent extraction methods*

*typically used to characterize and define brC*, as well as to quantify the amount of non-BC LAC present in a sample; whereas *tar brC* would *be detected by many of the methods used to quantify BC* (Corbin et al., 2019). Consequently, even simple quantities used to characterize BC such as mass-specific light absorption (i.e., MAC) may be biased in the presence of tar brC (Mason et al., 2018). Second, the physical characteristics described above would result in an environmental fate different to that of soluble brC, including slower reaction rates (Reid et al., 2018) and accumulation on snow and ice surfaces.

Of course, the properties specified here are continuous, and materials with intermediate properties between tar brC and soluble brC may be observed in the atmosphere. The potential existence (and potential atmospheric importance) of particles with intermediate properties between tar brC and soluble brC requires caution and the careful interpretation of data, but does not supersede the importance of distinguishing between tar brC and soluble brC for the reasons noted above. Examples of less-carbonized tarballs include the hygroscopic wildfire tarballs described by Hand et al. (2005) and the soluble laboratory

tarballs described by Li et al. (2019). Studies must therefore carefully characterize the LAC types discussed here, using a combination of techniques as necessary (Adler et al., 2019; Corbin et al., 2019), and keeping in mind the response of different techniques to tar brC of varying levels of carbonization. Since tar brC is of particular importance for its light-absorbing properties (Adachi et al., 2019), techniques which are biased towards more-absorbing tar brC should be favoured over those which are biased towards less-absorbing tar brC. The SP2 may represent a useful technique in the former category, as discussed in the

Introduction.

## 2.2 Definition

*"Tar brC" is the polymerized and potentially carbonized solid resulting formed by the thermal or chemical transformation of low-molecular-weight organic molecules. Tar brC is the material comprising "tarballs", and has been observed in wildfire and residual-fuel emissions.* As tar brC forms through continuous processes, materials of intermediate degrees of polymerization or carbonization may exist and have been observed in wildfire smoke.

This definition is based on the above review of the atmospheric and materials-science literature. The remainder of this subsection places the above revised definition of tar brC in the context of previous definitions.

The term "tarballs" was introduced by Pósfai et al. (2004) to describe the spherical, amorphous-carbon particles found in biomass burning smoke which were stable under the electron beam of a transmission electron microscope. This definition is inapplicable to studies which have not used electron microscopy, which cannot assert stability under an electron beam and may not be able to assert sphericity. In addition, later work showed that "tar ball aggregates" may also exist (Hand et al., 2005; Girotto et al., 2018). We therefore prefer the chemically-based definition given above.

An alternative definition of tarballs as aged or processed primary particles from biomass burning has been proposed (Pósfai et al., 2004; Tóth et al., 2014; Sedlacek III et al., 2018), based on the apparent formation of tarballs during atmospheric processing (Sedlacek III et al., 2018), although tarballs have also been observed in unprocessed plumes (Pósfai et al., 2003). We propose that this definition would be more useful and less ambiguous if inverted. Rather than restricting the term tarballs to processed primary particles (a definition which relies on knowledge of particle history), we recognize that atmospheric processing may form tarballs, but define tar brC based on its material properties. This definition also accounts for the fact that not all aged primary biomass combustion particles are tarballs.

This inclusive, material-based definition avoids the unnecessary confusion of requiring several separate names for the tarball-like particles measured in unprocessed wildfire plumes (Pósfai et al., 2003, 2004; Semeniuk et al., 2006; Adachi and Buseck, 2011; China et al., 2013); in laboratory biomass smoke (Vernooij et al., 2009; Chakrabarty et al., 2010; Adler et al., 2019); in dry-distilled, heat-treated laboratory biomass (Tóth et al., 2014; Hoffer et al., 2016a; Li et al., 2019); in fresh marine-engine exhaust (Corbin et al., 2019; Jiang et al., 2019); or in atmospheric air masses of unmeasured or unreported photochemical age (Niemi et al., 2006; Hand et al., 2005; Tivanski et al., 2007; Alexander et al., 2008; Zhu et al., 2013). As all of these particles possess similar material properties, a single name, tar brC in general and tarballs for individual spheres, is most appropriate for them. More-specific names, such as biomass tarballs and residual-fuel tarballs, may be used as appropriate.

## 3 Methods

### 3.1 Technical description of the Single Particle Soot Photometer (SP2)

The single particle soot photometer (SP2; Droplet Measurement Technologies, CO, USA) is designed to quantify the mass of BC in single particles by laser-induced incandescence (LII, Stephens et al., 2003; Schwarz et al., 2006). Particles exiting a nozzle at near-atmospheric pressure are guided by a sheath flow through into a continuous-wave, intracavity, 1064 nm Nd:YAG laser over

the course of roughly $20\,\mu$s. If particles are heated to incandescent temperatures by this laser, the resulting incandescent light $I(t)$ is measured by broadband (350–880 nm) and narrowband (350–550 nm) detectors; the broadband signals are normally used

due to their greater sensitivity. Elastic scattering of the laser radiation $S(t)$ is measured by a second detector. A third detector also measures light scattering, but using a four-element avalanche photodiode with the polarity of two elements reversed, such that the measured signal crosses through zero when a particle reaches a specific physical location relative to the detector. This so-called "split detector" allows the absolute location of a particle in the laser beam to be unambiguously identified, by using the scattering signals of non-evaporating particles as a measurement of the beam profile (Gao et al., 2007; Laborde et al., 2012a). In

the SP2 used during this study, data were acquired from all detectors every $0.4\,\mu$s. All data in this study were analyzed using the PSI SP2 Toolkit, version 4.114, to which the novel features discussed below were added.

The SP2 scattering detector was calibrated by reference to polystyrene latex sphere standards of diameter 269 nm. The incandescence detectors of the SP2 were calibrated using mass-selected rBC particles with mass-specific incandescence responses similar to diesel-engine rBC (Alfa Aeser Inc., FS, Lot #FS12S011). As LII signals are influenced by the material

properties of the rBC (Laborde et al., 2012b; Michelsen et al., 2015), it should be kept in mind that the mass calibration for different materials (such as the rBC in soot or tar) may be different.

The SP2 detects rBC cores with mass (or volume-equivalent size, considering a void-free material density of $1800\,\mathrm{kg\,m^3}$) from $\sim 0.7\,\mathrm{fg}$ ($\sim 80\,\mathrm{nm}$) to $\sim 200\,\mathrm{fg}$ ($\sim 600\,\mathrm{nm}$). Smaller rBC particles can also be detected, although with reduced counting efficiency (Laborde et al., 2012c). For canonical soot BC, the integrated mass obtained by SP2 has been validated as accurate

by multiple independent studies over more than a decade (Slowik et al., 2007; Kondo et al., 2011; Laborde et al., 2012c). A substantial amount of work has also employed the SP2 to investigate the internal mixing or "coating" of rBC with volatile materials (Liu et al., 2017, and references therein). Further details of the SP2 analysis of the present data sets are presented in Corbin et al. (2018b).

At least two conditions must be met for particles to incandesce in the SP2. First, they must experience a substantial net

heat input from the 1064 nm laser (over a period of $20\,\mu$s, due to the particle velocity through the laser). This corresponds to a minimum required MAE, so that the heating rate exceeds conductive, evaporative, and other cooling rates (Michelsen et al., 2015; Bambha and Michelsen, 2015). In the SP2, the major cooling mechanism is conductive heat transfer (Bambha and Michelsen, 2015), which has been reported as limiting its ability to detect spark-generated carbon nanoparticles (Gysel et al., 2012). The heating rate will depend most strongly on the material properties of the particle; these properties may themselves be

influenced by the heating process if annealing occurs (Michelsen et al., 2015; Sedlacek et al., 2018). Second, particles must be refractory up to $\sim 3000\,\mathrm{K}$, so that the corresponding incandescence is detectable (Schwarz et al., 2006). Particles types which meet these conditions include canonical soot BC (Schwarz et al., 2006) and metal-containing particles such as dust (Moteki et al., 2017) and volcanic ash (M. Gysel, unpublished data). Particle types which do not meet these conditions include canonical non-absorbing materials and tar brC. Non-absorbing materials (such as volatile organics, sulfates, and nitrates) will not absorb

the SP2 heating laser and are not refractory. Tar brC may absorb the SP2 heating laser, but is generally only refractory to about 1000 K (Corbin et al., 2019). Other forms of brC neither absorb substantially above $\sim 500\,\mathrm{nm}$ (Kirchstetter et al., 2004; Laskin et al., 2015; Moschos et al., 2018) nor are refractory (Lack et al., 2012).

## 3.2 Test data set

The test data set used here is a subset of the experiments described in previous publications (Corbin et al., 2018a, b, 2019) and summarized in the introduction. The subset of the data set used here corresponds to a single marine engine operated on the same heavy fuel oil (HFO) at the same engine load, but with the engine tuning parameters varied such that the mass fraction of tar brC relative to tar brC plus soot BC was either negligible (no tar) or $> 0.75$ (dominated by tar) (Corbin et al., 2019). Here, we refer to these different conditions as the "tar-free case" and the "tar-rich case", respectively. We emphasize that the engine produced tar-rich aerosols at low engine loads ($< 25\%$) when operated normally.

Tarballs were verified as present in this data set using electron microscopy (Corbin et al., 2019), and were quantified by a combination of techniques. Since those techniques were not specific to spherical particles, and since tarballs may have existed as agglomerates with other material, we have used the term "tar brC" to refer to the material (as discussed in Section 2).

We defined the relative importance of BC or tar in the samples according to the Absorption Ångström Exponent (AAE) calculated from the wavelength pair $\{370, 950\}$ nm. This AAE(370,950) was close to 1.0 for the tar-free case and $\sim 2.0$ for the tar-rich case. In the tar-rich case, the mass fraction of tar relative to tar brC plus soot BC was $> 0.75$, and over half of the total light absorption at 950 nm was due to tar and not soot BC. For the tar-rich case, the rBC/EC mass ratio was 0.18. In contrast, for the tar-free case, the rBC/EC ratio was 0.97 and there was no evidence of non-BC light absorption.

In addition to tar brC and soot BC, a substantial mass fraction of the emissions from this engine consisted of volatile PM. Specifically, approximately 50% of the emitted PM mass consisted of volatile organics (Eichler et al., 2017; Corbin et al., 2018a) and approximatedly 25% consisted of sulfates (Corbin et al., 2018a). These relative proportions changed with engine load; in particular, volatile organic PM mass increased with decreasing engine load. As noted above, the mass ratio of tar brC to soot BC varied from zero to $> 0.75$.

In the analysis herein, the tar-free case is used as a control case while the tar-rich case is used to identify features unique to tar particles. The two cases were measured on the same day using the same sampling configuration with no changes to the SP2. Day-to-day variability, engine-load dependencies, and other features of the engine emissions have been discussed previously (Corbin et al., 2018b, 2019).

## 3.3 Analysis methods

Our analysis methodology employs two fundamental quantities, the calculation of which are described in this section and summarized in Table 1.

### 3.3.1 Time-resolved scattering cross-section $C(t)$

To quantify tar evaporation in the absence of incandescence, we calculated the time-resolved partial scattering cross-section $C(t)$ of all particles in the test data sets. This procedure has been described in detail by Laborde et al. (2012b) and is illustrated in Figure 1. Briefly, the SP2 split detector is used to define the absolute position of particles in the SP2 laser. The effective laser beam shape is then determined as the median of all measured non-incandescing particles above a user-selected noise threshold.

(We inspected the data to ensure that most non-incandescing particles were also non-evaporating.) Calibrated scattering signals are then normalized to this idealized beam shape to obtain $C(t)$.

Above the noise threshold, $C(t)$ is a constant for non-evaporating particles. For evaporating particles, $C(t)$ decreases due to the decreasing particle volume and, potentially, changing refractive index of the particle (Moteki and Kondo, 2008; Laborde et al., 2012b). We therefore used the change in $C(t)$ as a method to identify evaporation and quantify the number fraction

of evaporating particles. We calculate the ratio of $C(t)$ at two selected times relative to the mode intensity of the laser beam, defined by $R(t_1, t_2)$ with the times given in terms of laser beam intensity:

$$R(-20\%, 20\%) = \frac{C(-20\%)}{C(20\%)} \tag{1}$$

where $C(-20\%)$ represents $C(t)$ at the time when an incoming particle reaches 20% of the laser-beam maximum for the first time. Conversely, $C(+20\%)$ represents $C(t)$ at the time when a particle reaches 20% of the laser-beam maximum for the

305 second time and has almost left the laser beam.

We note that the PSI SP2 Toolkit has long used $C(t)$ for so-called "leading-edge only" (LEO) analysis (Gao et al., 2007). The calculation of $C(t)$ described above and by Laborde et al. (2012b) is equivalent to a LEO fit of an empirical beam shape function (rather than a prescribed function such as a Gaussian, and without the constraint of fitting to the leading edge). The results presented below are therefore an extension of a previously validated and published approach. We also note that our method

for to detecting evaporating tar particles bears some similarity to the method of Moteki et al. (2014) for detecting attached BC particles.

### 3.3.2 Scattering cross-section at incandescence, $C(t_o)$

To investigate $C(t)$ after the evaporation of volatile material, we define $C(t)$ at the time of incandescence as $C(t_o)$. As was also the case in earlier versions of the PSI SP2 Toolkit, $C(t_o)$ was defined as the scattering signal which occurred just prior to the

315 onset of incandescence. The condition "just prior to" is necessary because the filter used in front of the SP2 scattering detector transmits a portion of the incandescence signal, and because rBC particles may swell during heating (Bambha and Michelsen, 2015). The $t_o$ was defined as $2.4\,\mu s$ seconds before $t_i$, the time of maximum incandescence signal. This is illustrated in Figure 1.

We note that earlier versions of the PSI SP2 Toolkit also retrieved $C(t_o)$ and used it to constrain the apparent refractive index of rBC, $m_{\mathrm{rBC}} = (n, k)$. Because the precise value of $m_{\mathrm{rBC}}$ is not well constrained and may vary between BC materials, a range

of possible values for $m_{\mathrm{rBC}}$ have been reported in the literature (Bond and Bergstrom, 2006). These values have been empirically observed to follow the approximate relationship $k \approx (n-1)$, as introduced by Bond and Bergstrom (2006) and discussed further by Moteki et. al. (2010). This empirical relationship is sufficient to constrain $m_{\mathrm{rBC}}$ to a single value, within the scope of the Mie approximation used herein. This method was used in Corbin et al. (2018b) to determine a best-fit $m_{\mathrm{rBC}}$ of $1.9 + 0.8i$ for this data set, which is similar to the value used by Laborde et al. (2012c) for propane-flame soot but smaller than the value of (2.26,

1.26) often used in SP2 data analysis (Moteki et al., 2010; Taylor et al., 2015).

### 3.4 Data filtering

The particle-detection events analyzed herein were filtered to remove events with fitted peak heights below a limit of detection (LOD) established by inspecting the smoothness of the measured mass distributions. A higher LOD was applied to events analyzed for $C(t)$. This higher LOD was defined at $C(-3\%)$, which is a standard reference condition for LEO analysis in the PSI SP2 Toolkit and reflects the earliest time at which the signal-to-noise ratio of $C(t)$ becomes acceptable. The LOD at $C(-3\%)$ was determined by inspecting a scatterplot of the diameters of non-incandescing particles retrieved at $C(-3\%)$ versus those retrieved at $C(100\%)$, the standard position.

Additional filters were introduced to remove events triggered by noise, events where peak fitting failed (for example, due to the coincidence of two particles within the laser beam and with one particle touching the edge of the acquisition window), and events where the detectors were saturated. The number fraction of particles removed by these filters was negligible. For $C(t)$ data, only particles with a valid split position could be used, which corresponds to a lower limit of approximately $160\,\mathrm{nm}$ in optical diameter (assuming $m = 1.5 + 0i$).

## 4 Results

In the following discussion we present results which provided useful evidence for the presence of unique signals from tar particles in the SP2. Appendix A describes some results which did not provide useful evidence.

### 4.1 Evaporating, non-incandescing particles

Figure 2 shows frequency distributions for $R(-20\%, 20\%)$ (Equation 1) for particles where no incandescence signal was observed. The figure shows data only for particles with optical diameter of $220 \pm 40\,\mathrm{nm}$ (assuming $m = 1.5 + 0i$) to remove noise from smaller particles and to simplify interpretation.

The $R(-20\%, 20\%)$ represents the fraction by which the scattering cross-section $C(t)$ of a particle decreased when exiting the beam (sampled at $20\%$ of the maximum laser fluence) compared to entering the beam (also sampled at $20\%$). Its value is close to 1 for a non-absorbing particle and $\ll 1$ for an evaporating particle. Values greater than 1 occur due to random error in $C(t)$.

For both the tar-free (control) and tar-rich cases, the vast majority of signals can be described by a gaussian fit as falling within the range $0.7 < R(-20\%, 20\%) < 1.3$. For the tar-rich case the fitted mean and standard deviation were $1.001 \pm 0.001$ and $0.279 \pm 0.002$ respectively. This standard deviation reflects a $27.9\%$ precision in the retrieval of $R(-20\%, 20\%)$. For the tar-rich case, a substantial but small fraction of the particles in Figure 2 (578 of 14766 or 3.9%) showed $R(-20\%, 20\%) < 0.5$, indicating substantial evaporation. The $C(t)$ profiles of all of these particles are shown in Figure 3. Note that the majority of particles in this sample were non-absorbing lubrication-oil particles in the sample, which cannot be quantified separately from non-evaporating tar. Therefore, the actual number fraction of tar particles which evaporate (relative to tar particles which do not evaporate nor incandesce) is likely to be significantly higher than the 3.9% given above.

By random inspection of the evaporating particles with $R(-20\%, 20\%) < 0.5$, we selected a representative example and plotted its time-resolved scattering signals in Figure 1d. The scattering cross-section $C(t)$ of this example particle begins at a plateau (indicating unchanging particle size and composition) before decreasing to a second plateau (indicating a second stable configuration), then ultimately evaporating completely. Not all evaporating and non-incandescing particles showed this secondary plateau, many showed only a continuous evaporation, some showed a partial evaporation and remained at a plateau (Figure 3). It may be speculated that these plateaus reflect the breakup of tar particles into more- and less-absorbing parts, with the more-absorbing part being completely evaporated and the less-absorbing part passing through the laser unchanged, in analogy to the particle breakup that is observed for heavily-coated rBC (Sedlacek et al., 2012; Dahlkötter et al., 2014; Moteki et al., 2014).

This evaporating, non-incandescing behaviour is unlike any we have previously observed in the SP2. Typically, non-incandescing particles will show a scattering profile similar to that of Figure 1a, corresponding to constant $C(t)$. Recall that the beam profiles shown in Figure 1 reflects the median of all non-incandescing particles, and therefore may be interpreted as illustrating the scattering profile of a typical non-incandescing particle.

Evaporating but non-incandescing particles may also be observed in the SP2 when sub-detection-limit rBC is internally-mixed with volatile material. This may result in core-shell particles, which were not observed in the tar-rich nor in the control data set. Alternatively, this may result in the "attachment" of soot particles to non-absorbing droplets, rather than core-shell morphologies. However, again, this should have been observed in our control data set but was not. Finally, the presence of sub-detection-limit rBC would have occurred simultaneously with the presence of larger rBC particles, so that some partially-evaporating and incandescing particles should have been seen (as observed for example by Moteki et al. 2014) if this possibility were significant. No particle breakup was obseved in our data; no incandescing particles were observed which had a detectable signal remaining at $C(20\%)$. Therefore, this potential cross-sensitivity can be excluded as affecting our analysis.

## 4.2 Evaporating and incandescing particles

We investigated the possibility that certain tar particles may incandesce in the SP2 due to laser-induced annealing. This possibility requires distinguishing incandescing tar from incandescing soot BC particles (and potentially internally-mixed tar–soot particles). This distinction could be made using a comparison of the scattering and incandescence signals for tar particles, via the ratio $C(t_o)/I_{\text{peak}}$, under the hypotheses that tar particles either (i) contain a substantial volume of refractory, non-incandescent material at the time of incandescence $t_i$ or (ii) possess a substantially different refractive index at $t_i$. Our data do not rule out the possibility that both hypotheses are true. These symbols were defined in Table 1. For this calculation, it is important to use $C(t)$ and not $S(t)$, because thickly-coated rBC particles penetrate deeper into the SP2 laser. This results in a higher $S(t)$ for the same $C(t)$, since deeper penetration into the laser corresponds to a greater photon flux incident on the particle. We note that the incandescing material in laser-annealed tar would have a significantly different morphology and possibly also molecular structure than soot BC, which would affect its incandescent properties (Moteki and Kondo, 2010). .

Figure 4 shows $C(t_o)/I_{\text{peak}}$ for the tar-rich and tar-free (control) cases, plotted as a joint-probability histogram of initial optical diameter. The initial optical diameters correspond to the diameter retrieved from $C(-3\%)$ assuming $m = 1.5 + 0i$ and

serves to indicate the approximate particle size prior to any evaporation. For soot BC, when the ratio $C(t_o)/I_{\text{peak}}$ is appropriately calibrated, it represents the ratio of optical diameter at $t_i$ to rBC-equivalent diameter at $t_i$. The ratio would then represent the slope of a plot of the rBC optical diameter (just prior to incandescence) against rBC mass-equivalent diameter (obtained from the incandescence signal), and in this scenario the ratio is constrained as equal to unity during BC-coating-thickness analyses when the appropriate calibrations are applied (Corbin et al., 2018b). For clarity we have therefore applied these calibrations to the data presented in Figure 4, although they do not apply to tar.

Figure 4a shows the $C(t_o)/I_{\text{peak}}$ versus diameter histogram for the tar-free case. An approximately constant $C(t_o)/I_{\text{peak}}$ is observed as a function of initial optical diameter, as highlighted by the dashed ellipse. Above 200 nm, $C(t_o)/I_{\text{peak}}$ begins to decrease, reaching a value of 0.9, which at least partially reflects the limitations of the Mie model used to calculate the optical diameter in our analysis. The region within this dashed ellipse reflects the SP2 response to soot BC. Inspection of the individual particles within the soot BC region showed the canonical SP2 response, as depicted for a representative particle in Figure 1b. This response involves a scattering signal $S(t)$ that decreases almost simultaneously with incandescence and a scattering cross-section $C(t)$ that drops rapidly after the onset of incandescence.

Figure 4b shows the $C(t_o)/I_{\text{peak}}$ versus diameter histogram for the tar-rich case. Here, in addition to the soot BC region, a second "cloud" of particles appears at higher $C(t_o)/I_{\text{peak}}$ and higher initial diameter, as highlighted by the solid circle on the figure. The circle is reproduced on Figure 4a to allow a direct comparison of the two data sets. Inspection of the individual particles within this "tar region" showed an anomalous SP2 response, as depicted for a representative particle in Figure 1e. Unlike the coated soot particle, no plateau in $C(t)$ is observed at incandescence, rather, incandescence occurs simultaneously with a continuously decreasing $C(t)$. After incandescence, $C(t)$ is zero. This behaviour indicates that a substantial amount of refractory material was internally mixed with the material which incandesced. The material which incandesced may have undergone chemical transformation during laser heating (laser-induced annealing as discussed above, Sedlacek et al. 2018) or may have been present prior to laser heating; we consider annealing more likely due to the homogeneous appearance of these tar particles in the electron microscope (as discussed in Corbin et al., 2019).

The area of Figure 4 containing incandescing tar particles is significantly greater than the area containing soot-BC particles (that is, the illustrative circle is larger). This indicates that tar particles showed a more variable ratio of non-incandescing to incandescing material, and/or that the incandescing material varied in degree of graphitization or annealing. This is the expected behaviour, considering that the annealing process involves the localized crystallization of graphitic domains following thermal decomposition, or heating-related internal tensions (Franklin, 1951). Such a phase transition would occur at variable times during laser heating, leading to a variable ratio between evaporated and annealed material, leading to a variable $C(t_o)/I_{\text{peak}}$ ratio.

The clear relationship between $C(t_o)/I_{\text{peak}}$ and initial optical diameter seen in Figure 4b rules out the hypothesis that this region reflects extremely-thickly-coated soot. While soot coatings ideally undergo complete evaporation in the SP2 prior to incandescence, extremely-thick coatings may result in particle breakup (Moteki and Kondo, 2007; Sedlacek et al., 2012; Dahlkötter et al., 2014). That is, the coating may fragment and generate a secondary particle large enough to generate a scattering signal in the SP2. This fragment particle would not be in thermal contact with the rBC core and would therefore generate a

stable $C(t)$ signal, which would be observed simultaneously with the $C(t)$ signal from the rBC core. The $C(t)$ of the fragment particle would therefore cause additional scattering at the time of incandescence, and shift the resulting signal towards higher $C(t_o)/I_{\text{peak}}$ values in Figure 4. Moteki and Kondo (2007) observed that graphite particles coated with oleic acid or glycerol did not undergo fragmentation until initial diameters of 400 nm or 600 nm, respectively. The fraction of fragmenting particles then

increased rapidly until virtually all particles fragmented at 500 nm or 650 nm, respectively. Since no such rapid transition is seen in Figure 4, and since our data set employed a tar-free control case, we can be confident that fragmentation did not play a role in our data set. Moreover, as mentioned earlier, we have verified by manual inspection of our data that the particles labelled "tar" in Figure 4 were not thickly-coated and that no scattering signal remained after incandescence.

Finally, we also present a coated soot-BC particle in Figure 1c in order to illustrate the behaviour of such particles in the

SP2. We emphasize that, unlike all other examples, this particle type was very rare in our data set, and is not representative of our data set, in which most BC was uncoated (Corbin et al., 2018b). This coated particle shows a $C(t)$ that decreases to a plateau prior to the onset of incandescence. This is the typical behaviour of coated rBC. The initial decrease indicates the evaporation of a volatile coating, and the plateau indicates continued heating of the now-uncoated rBC up until the onset of incandescence at $\sim 3500\,\text{K}$. The onset of incandescence corresponds to a slight increase in $C(t)$, which must be interpreted

in the context of heat-induced swelling and interference of the incandescence signal at the scattering detector (Bambha and Michelsen, 2015). Overall, the scattering and incandescence profile of this coated soot BC particle is clearly distinct from the profiles of tar. We note that some particles with extremely-thick rBC coatings or coagulated rBC-droplet morphologies may not display the above-mentioned plateau due to breakup during evaporation (Sedlacek et al., 2012; Moteki et al., 2014), but such particles also display a $C(t)$ significantly greater than zero after evaporation, and, as mentioned above, were not observed in our

data set.

## 5    Discussion

### 5.1    Laser annealing of tar to form rBC within the SP2

The results presented above show that marine-engine tar particles absorb the 1064 nm SP2 laser with sufficient efficiency to evaporate. In some cases, incandescence accompanied this evaporation, which was attributed to partial laser-induced annealing.

That is, we believe that the rapid laser heating allows part of the initial tar particle to rapidly anneal, forming graphitic domains which are refractory enough to incandesce similarly to rBC, and which are of sufficient volume for the incandescence signal to be detected by the SP2.

While this incandescence-via-partial-annealing phenomenon was directly demonstrated by Sedlacek III et al. (2018) using nigrosin and laboratory-generated tar particles, in our study there is a possibility of rBC pre-existing as an internal mixture with

tar. We consider this possibility extremely unlikely based on the unique formation mechanism of tar compared to soot BC, and the fact that tar particles have universally been observed as internally homogeneous in the literature (Section 2).

A competing hypothesis to the partial annealing hypothesis is that the incandescing tar particles were actually coagulated tar–soot-BC particles. We reject this hypothesis, because of the observed late incandescence (in terms of time spent in the laser

beam) of tar particles. Coagulation would not result in late incandescence; at worst, it would lead to earlier incandescence due to reduced conductive cooling.

It is important to realize that laser annealing is dependent upon the laser intensity within the SP2 cavity, as systematically demonstrated by Sedlacek et al. (2018). Sedlacek et al. (2018) also showed that the minimum laser power required to induce detectable annealing depends on the starting material, and decreases if a given starting material is heated in a furnace prior to measurement by SP2. Given that tar brC is always produced in high-temperature systems (whether the source is a marine engine or a wildfire) where such prior heating is likely to be variable, additional systematic work will be needed to establish a reliable protocol for the SP2-based measurement of tar brC particles. Future studies may find it helpful to exploit furnace pre-treatment as an option for enhancing the ability of the SP2 to detect tar.

## 5.2 Detection of tar brC by other real-time instruments

If tar particles absorb 1064 nm light, then they should be measurable by other techniques which employ 1064 nm lasers, such as the Soot-Particle Aerosol Mass Spectrometer (SP-AMS) (Onasch et al., 2012) and pulsed laser-induced incandescence (pulsed LII) instruments (Michelsen et al., 2015). Corbin et al. (2019) explored the response of the SP-AMS to the same tar particles discussed in the present manuscript and found no substantial difference between the mass spectra of tar-containing and tar-free samples. This lack of difference may be due to the different conditions that particles experience within the SP-AMS (including lower pressure, shorter beam exposure times, and different laser power densities, as mentioned in the introduction), or due to the fact that the SP-AMS used in our study did not obtain the single-particle measurements that allowed the SP2 to differentiate between tar and lubrication-oil-related particles. Future work should explore the response of pulsed LII instruments to tar.

When not agglomerated with other particles (Pósfai et al., 2004), tar particles are unique in being refractory, spherical, and strongly light-absorbing (Corbin et al., 2019). It is would therefore also be possible to characterize tar particles in real time by heating an aerosol sample to remove non-refractory material (leaving only soot BC, tar brC, and potentially char BC) before using a combination of two different particle classifiers to produce an aerosol composed primarily of tar. This approach has been demonstrated by Adler et al. (2019), although those authors did not refer to their particles as tar particles, as noted above.

## 5.3 Relative number of incandescing and non-incandescing tar particles

The number fraction of evaporating tar particles observed in our data set was 3.9% at $220 \pm 40$ nm. This fraction is biased by the fact that the majority of particles in our sample were lubrication-oil related (Corbin et al., 2018b). The number fraction of incandescing tar particles was 1324 of $2.62 \times 10^5$, or 5.1%. For a given optical diameter, tar particles generated much smaller incandescence signals than soot particles (Figure 4), so that the actual bias in SP2-determined rBC mass concentrations due to tar incandescence was $\ll 5.1\%$. Considering that the mass of tar brC was threefold greater than the mass of soot BC in our measurements (Corbin et al., 2019), this bias is negligible relative to the typical 15% accuracy of an SP2 mass calibration (Laborde et al., 2012c; Taylor et al., 2015).

It is important to note that the statistics reported here are dependent on the history of the particles studied, including the time spent at high temperatures within the engine as well as the SP2 laser power (see Section 5.1).

## 5.4 Comparability with tarballs described by other studies

It cannot be overemphasized that the material referred to as tar or "tarballs" is partially-graphitized, amorphous carbon. There is no therefore no well-defined molecular structure for tar, and a given tar sample may lie at some point along a continuum of carbonization (Corbin et al., 2019). While the same is also true of soot BC (Minutolo et al., 1996; Vander Wal et al., 2014), the literature indicates that the range in degree of carbonization of BC emitted by common combustion sources (Bond and Bergstrom, 2006; Zangmeister et al., 2018) is narrower than the corresponding range for tar (Corbin et al., 2019).

Care must therefore be taken when extrapolating the present results to other studies. Certain tar samples may be less carbonized (and less likely to undergo laser-induced evaporation) or more carbonized (and more likely to undergo laser-induced incandescence) than our samples. This includes residual-fuel tar brC samples produced by different engines or different combustion systems, as well as biomass-burning tar brC. Based on the wavelength-dependence of absorption of tar in our samples reported and placed in the context of literature by Corbin et al. (2019) (AAE of about 3), we believe that our tar samples were of a typical degree of graphitization. Future studies should explore the possibility of modulating the SP2 laser power (Sedlacek et al., 2012, 2015), to provide additional information by which tar and soot BC may be distinguished.

More generally, in all studies on LAC particles, it is essential to report not only a name for the particles being studied, but as many properties as is practical, and as are justified by the novelty of the particle source. These properties include light absorption efficiency (MAC), wavelength dependence (AAE), morphology (which directly influences light absorption), volatility and solubility in relevant solvents. Such a comprehensive analysis allows a sample to be placed along the continuum of carbonization which represents the soluble brC to tar brC continuum, or along the continuum of graphitization which represents the maturity of soot aggregates.

## 6 Conclusions

We investigated the response of the SP2 to near-infrared-absorbing, refractory (to about $1000\,\text{K}$) carbonaceous particles ("tar") using a data set in which the presence of tar has been demonstrated, and a control data set in which such particles were absent (Corbin et al., 2019).

By inspecting the time-resolved scattering cross-sections $C(t)$, we found that tar particles can be observed as evaporating but non-incandescing particles in the SP2. Some tar particles also incandesced, either due to laser annealing or possibly due to chemical heterogeneity of the material being referred to as tar. These incandescent tar particles were clearly distinguishable from soot BC according to the ratio of scattering-at-incandescence to incandescence signals. This ratio was a factor of 1.2 to 2.0 greater for tar than for soot BC and much more variable. This high degree of variability would be expected if the incandescent material in these particles formed via the localized crystallization of graphitic domains during laser-induced annealing, such that the molecular composition of the incandescing tar particles may vary significantly.

In our data set, we identified 578 and 1324 particles as non-incandescing or incandescing tar particles, respectively. Assuming that the probability of false-negatives is similar for these two statistics (in other words, assuming that our different methodologies were not more sensitive to either incandescing or non-incandescing tar), this indicates that about 70% of tar particles produced

incandescence signals in the SP2. The number fraction of evaporating, non-incandescing particles (evaporating tar brC) relative to all tar particles or simply relative to all particles is not reported due to the presence of significant amounts of lubrication-oil related particles (Eichler et al., 2017), which would bias this number fraction low by an unconstrained amount. Future work should employ sample pretreatment (thermal denuding), laboratory-generated tarballs, or morphology-based classification to more accurately estimate what fraction of tar particles can be expected to evaporate in the SP2 laser.

The analysis presented here shows that an SP2 equipped with a split detector is capable of detecting tar. This makes the SP2, to our knowledge, the only high-throughput technique which has the potential capability of distinguishing tar from soot particles or soluble brC. It remains undetermined whether or not the SP2 signals are useful for the quantification of tar mass or number fractions. Based on the fact that our tar particles had optical properties similar to those reported in other studies (discussed in Corbin et al., 2019), we estimate that our tar particles are of a typical degree of carbonization, such that other tar-containing samples should display similar behaviour to that observed herein. Future work should also explore the possibility of modulating the SP2 laser fluence in order to exploit differences in the absorption efficiencies of tar and soot, while keeping in mind that the material referred to as tar lies on a continuum between amorphous carbon and highly graphitic carbon, such that certain samples will absorb 1064 nm light more effectively than others, while also most likely being more refractory (Corbin et al., 2019). The techniques used herein may be useful for the future identification of the presence or absence of tar in a sample of unknown composition.

**Author contributions**

JCC and MGB conceptualized the study; JCC performed the research and drafted the manuscript with input from MGB; MGB developed the original SP2 analysis code which JCC further developed; and MGB acquired funding.

*Acknowledgements.* This work was supported by the ERC grant ERC-CoG-615922-BLACARAT and by Natural Resources Canada under the Program of Energy Research and Development (PERD). We thank our previous coauthors (Corbin et al., 2018a, 2018b, 2019) for their contributions to generating the data sets used here. We acknowledge helpful input from two anonymous reviewers.

**Table 1.** Table of symbols used in the text.

| Symbol | Meaning |
| --- | --- |
| $t$ | Time spent by a particle in the SP2 laser beam |
| $S(t)$ | SP2 scattering signal |
| $C(t)$ | Scattering cross-section corresponding to $S(t)$ |
| $C(-3\%)$ | Scattering cross-section at -3% of laser maximum |
| | as a particle enters the SP2 laser beam |
| $I(t)$ | SP2 incandescence signal |
| $I_{\text{peak}}$ | $= I(t_i) =$ SP2 incandescence signal at peak |
| $t_o$ | Time just before onset of incandescence |
| $t_i$ | Time of maximum incandescence |
| $R(-20\%, 20\%)$ | Ratio of $C(t)$ at two different $t$ (Eq. 1) |

## Appendix A: Diagnostics which did not differentiate between tar and the control data set

We attempted to identify evaporating particles using a number of different statistics, with the goal of identifying a parameter which was sensitive to evaporation without requiring the split detector. One motivation for a split-detector-free method is that

the new model of the SP2, SP2-XR, does not contain a split detector. Using the region of the scattering signal identified as a peak by the PSI SP2 toolkit, we calculated the full-width-half-maximum (FWHM), the ratio of FWHM to full width, the peak skewness, the mean absolute difference (MAD) between either half of the peak, and the Kolmogorov-Smirnov statistic. These statistics were evaluated by manual inspection and by comparison to $R(-20\%, 20\%)$.

Manual inspection of the MAD suggested that this statistic successfully isolated tar particles, however, in terms of probability

density functions, the MAD for the tar-rich case was not different from the tar-free case. Manual inspection of the FWHM of the incandescence peak also suggested a difference for the tar-rich case, but further analysis showed that this difference was due to tar particles penetrating deeper into the laser beam, and therefore experiencing higher heating rates at incandescence. We also explored the use of the Moteki and Kondo (2008) approach, which is designed to provide the same information as the split detector from the raw data of the scattering trace, to identify evaporating, non-incandescing particles, but did not identify

conditions where this method was successful. Based on plots similar to Figure 3 but for various subsets of particles, we believe that alternative approaches must be explored, such as the machine learning approach introduced by Lamb (2019). Alternatively, future work may be able to distinguish tar-containing particles without a split detector if additional instrumental parameters, such as the laser fluence, are varied, or, more simply, by comparing the number of non-incandescing but refractory particles measured after a thermal denuder.

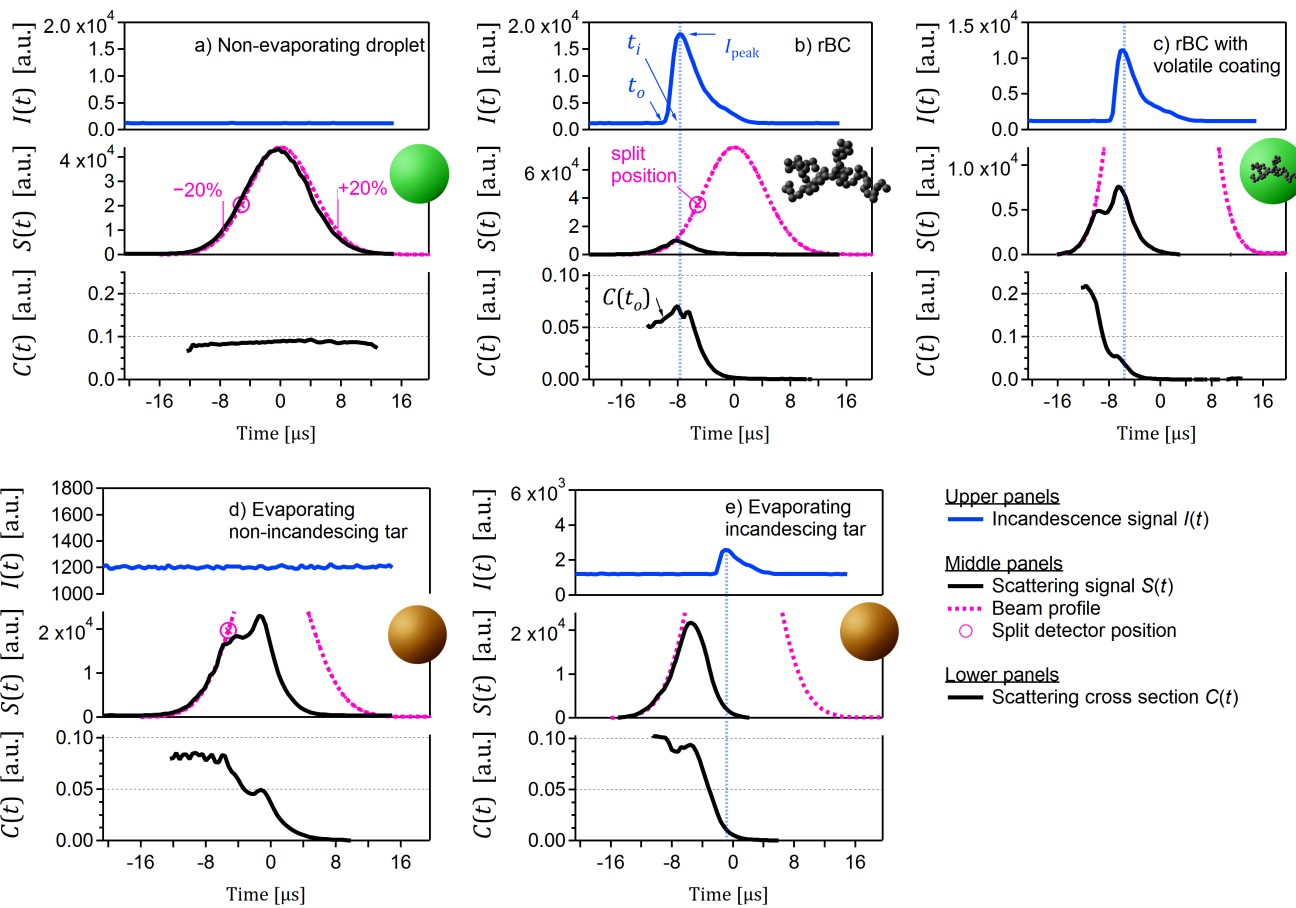

**Figure 1.** Particle types observed in this study. These examples were selected to be representative of the trends shown in subsequent figures. a) Typical non-absorbing and therefore non-evaporating particle. (Likely lubrication oil / sulfate mixture.) The labels $-20\%$ and $+20\%$ indicate fractions of maximum beam intensity. b) Typical soot BC particle. c) Atypical coated soot BC particle (rare in this data set, selected for illustration only). d) Typical evaporating but non-incandescing tar particle. e) Typical evaporating and incandescing tar particle. Note the difference between position of maximum incandescence (vertical blue lines) and position of stable $C(t)$ in panels c) and e), as highlighted by the vertical blue lines. Note also that the ordinate scales vary in order to highlight key features.

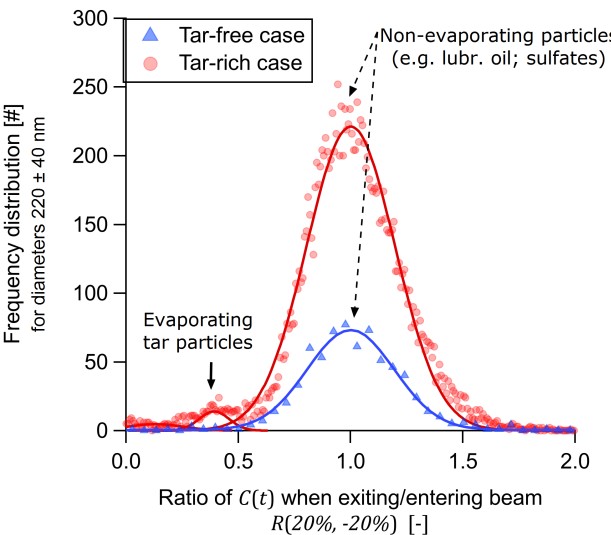

**Figure 2.** Frequency distribution of the ratio of scattering cross-sections, $R(-20\%, 20\%)$ for non-incandescing particles of diameter $220 \pm 40\,\text{nm}$. Triangles and circles represent engine conditions where the mass fraction of tar brC relative to tar brC plus soot BC was zero or 0.75, respectively. Ratios were calculated by dividing the second scattering cross-section measurement at 20% of maximum beam fluence (particle exiting the SP2 laser) by the first (particle entering the SP2 laser). Random error in $R(-20\%, 20\%)$ was modelled by gaussian fits, which indicated that about 3.6% of tar-rich case particles evaporated to less than half of their original cross-section. Note that this 3.6% reflects the fact that the majority of particles in this sample were non-absorbing lubrication-oil particles. In the tar-free case, no non-incandescing particles evaporated.

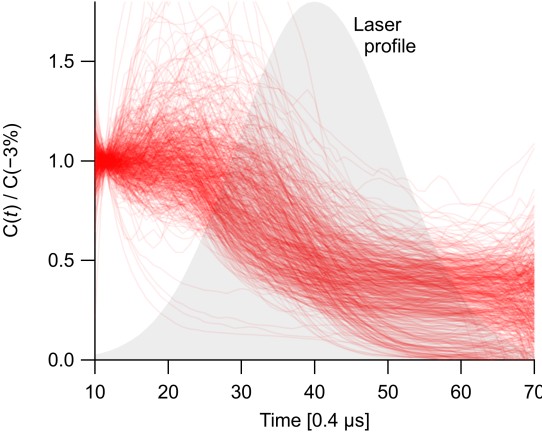

**Figure 3.** $C(t)$ for all particles with $R(-20\%, 20\%) < 0.5$ in Figure 2, normalized to $C(-3\%)$ (which is normally using for coating-thickness analysis). Each transparent red line represents $C(t)$ for a single particle, as was also shown in the lowest panel of Figure 1d.

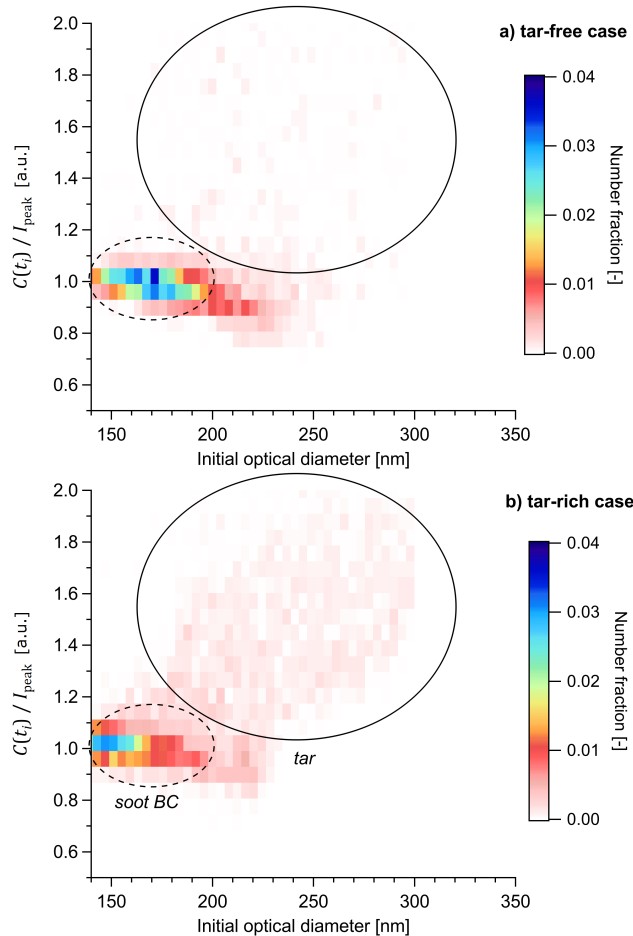

**Figure 4.** Incandescing tar particles distinguished from soot BC by their scattering properties in the SP2. The abscissa shows the optical particle diameter prior to evaporation, calculated from $C(t_i)$ with $m = 1.9 + 0.8i$. The ordinate shows the ratio of optical-equivalent and incandescent-equivalent particle diameters, $C(t_i)/I_{\text{peak}}$, calculated with the optical properties of soot BC. The colour scale shows the number fraction of particles in each bin (that is, a normalized joint probability histogram). Note that $C(t_i)/I_{\text{peak}}$ employs the optical diameter at time of incandescence $t_i$ and therefore represents the optical diameter of the refractory component of the particle. Therefore, for soot BC $C(t_i)/I_{\text{peak}} \approx 1$ by definition, although deviations below unity are observed for larger particles where light-scattering transitions from the Rayleigh regime to the geometric regime (Moteki and Kondo, 2010). Thickly-coated soot BC particles are also expected to fall on the $C(t_i)/I_{\text{peak}} \approx 1$ line since volatile coatings evaporate before $C(t_i)$ is measured, as discussed further in the text. In contrast, $C(t_i)/I_{\text{peak}}$ is very different from unity in the presence of particles which contain non-incandescing but refractory material.

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
