# Peer review of "Detection of tar brown carbon with the single particle soot photometer (SP2)"

_Atmospheric Chemistry and Physics, 2019_

## Referee Comment (RC1) · Anonymous Referee #1 · 18 Aug 2019

The current manuscript builds upon recent findings that the SP2 can induce some NIR light absorbing organic aerosol particles (commonly referred to as brown carbon or BrC) to char and subsequently be detected as refractory black carbon in the SP2. Specifically, the present manuscript examines tar brown carbon particles (referred to as "tar brC") that are produced by a marine engine running on heavy fuel oil. Using both the time-dependent scattering and incandescence channels of the SP2, the authors report that tar BrC annealed by the SP2 laser, and then detected as "rBC", can be distinguished from primary emission rBC from incomplete combustion. This new analysis methodology provides a framework that will expand the utility of the SP2 to include the subclass of BrC particles that absorb at the lasing wavelength of the SP2 (1064 nm). The methodology and analysis presented are sound and the paper is writ-

ten well. Publication is certainly warranted, once the authors address a concern on material definition and clarify a nomenclature issue as highlighted below.

With respect to material definition, a core statement that the authors make, and one that traces back to the lead author's 2019 Climate and Atmospheric Science paper, is that tar ball material is insoluble (e.g., cited on page 2, line 27 and again page 3, line 89-90 in present manuscript). The authors are encouraged to look take a second look at the closing section in Hand et al., 2005 where these authors state (page 12; ". . ..ESEM experiments of particle hygroscopicity indicate that tar balls observed during YACS are water soluble at high relative humidity (RH > 83%), albeit to a much lower degree than inorganic salt particles and with no distinct deliquescence point." Additionally, this reviewer wishes to draw attention to the recent work of Li et al., (ACP, 19, 139-163, 2019) where the presence of both polar and non-polar TBs (e.g., water soluble and non-soluble components) are reported on and discussed. The observation that wildfire TBs are water soluble and that some laboratory-generated TBs can be as well, suggest that care should be exercised with the application of generalized labels such as that used in the present manuscript. While the tar brC particles investigated in the present study may very well be completely insoluble, the reported presence of water-soluble TBs suggests that there are in indeed two classes of "tar-like" particles and further suggests that some reconciling is in order. The fuel source for a marine engine is very different from that available for wildfires - not to mention the emission sensitivity to burn conditions in the latter - so perhaps we should not be too surprised that there could be some fundamental differences of specific particle types within this subclass of brown carbon particles. The authors need to address this and adjust their material definition argument accordingly. If indeed biomass burning tar particles are different from fossil fuel tar particles, how generalizable are the conclusions about tar brC behavior in the SP2 laser?

This reviewer is confused by the presence of a non-incandescing distribution for the "soot BC case" (blue line) in Figure 2. In section 3.2, the authors reference a "pure-BC

case" as a "control" to help identify the unique features of the tar particles, yet in the figure, they cite "soot BC case". It is not clear whether the authors are referencing two different "BC" controls or are using two labels for the same material. My concern is that a "pure-BC case" should be composed of solely rBC particles that would all incandesce in the SP2 and that would necessarily lead to a non-Gaussian scattering signal distribution for in Figure 2. Additionally, the traces presented in Figure 2 are presumably for non-incandescing particles, which, again, for a pure-BC case, no pure scattering signals should be present. In short, clarification is in order. I believe the solution is straightforward enough. As this reviewer understands the manuscript text, the "soot BC case" are those particles generated when the marine engine is running under a specific operation condition and hence serves as a control on the particle-type produced and believes that their reference to "pure-BC case" is the same material. If correct, please clarify this nomenclature and stick to one label for your control material.

Other specific issues:

Page 1, lines 5 -6. The authors write "….tar brC results in unique SP2 signals due to a combination of complete or partial evaporation, with no or very little incandescence. Approximately 70% of tar particles incandesced." The juxtapositioning of these two sentences is very awkward. The first seems to say that there is very little to no in-candescence while the follow on sentence says that 70% of the particles incandesced. Which is it? Please reword.

Page 4, line 95, 96. The authors are encourage to examine (and possibly reference, if they think it is relevant) the recent paper by Li et al., (ACP, 19, 139-163 (2019)) where the optical properties of laboratory-generated TBs were investigated and refractive indices derived.

Page 6, line 183-185. The ∼3000K refractory requirement is dictated by the bandpass filters used in the commercial-grade SP2. A different set of bandpass filters would enable a lower temperature blackbody to be detected. (Admittedly, this reviewer is

being a bit pedantic here, but still. . ..)

Page 18, Figure 4. While the authors discuss how different thickly-coated rBC particles would look cast on a similar plot as shown for two cases in Figure 4, this reviewer wonders whether it would be useful to have a third panel on this figure for thickly-coated particles. As a picture is worth a thousand words, so too is a plot worth a thousand arguments and thus might prove a useful visual comparison for those readers who are not well acquainted with the SP2 mixing state analysis and data products. This is certainly not critical, but rather a suggestion, and thus at the authors discretion.

————————————————————

---

## Referee Comment (RC2) · Anonymous Referee #2 · 27 Aug 2019

Summary:

The authors present evidence that a material they identify as tar brown carbon (tar BrC) has a unique signal in the SP2 (evaporation but little/no incandescence) based on observations of emissions from ship engines. They use the time dependent scattering channels and broad-band incandescent channel in the SP2 to provide a method to identify these tar BrC particles. These observations provide additional evidence of further utility of the SP2 in identifying other aerosol types than rBC, and provides a new on-line measurement technique for a specific class of brown carbon that absorbs at the wavelength of the SP2 laser. The analysis approach for identifying the tar BrC is clearly described, and the authors also thoroughly investigate an alternative explanation for the tar BrC associated with low incandescence (if it might instead be thickly coated

rBC). The writing is very clear and the manuscript is logically written. This paper will be a valuable addition to the literature once a few minor points have been addressed.

General Comments: I would like to see more description of the data sets and observations used in the study; although this paper referenced Corbin et al. 2019 (where these measurements were described in more detail), it would make this work more useful as a stand-alone paper to have some additional details of the observations given here. These details would also help the reader better understand how these specific observations of aerosols from ship engine emissions might relate to other aerosol sources, such as biomass burning. Additionally, the authors have pointed out that the in-SP2 annealing that has been observed in Sedlacek et al. 2018 but not in Moteki and Kondo 2008 may be due to differences in experimental procedures; thus, a greater description of the SP2-specific calibrations in this work would be warranted.

I share some of the concerns of Referee #1, that specifically referring to these aerosols detected by the SP2 as "tar BrC" may be mis-leading, given the significant literature surrounding tar balls, and the fact that the aerosols in this current study are identified solely based on their observed signals in the SP2. However, as the authors have taken the time to clarify this in Section 2, I do not have major concerns about this terminology.

Specific Comments: p.7 Section 3.2. Please clarify whether the SP2 operating conditions were the same during observations of both the pure-BC case and the tar-rich case, as this would impact the interpretation of the experiments.

p.8 line 250-252. Clarify if 160 nm is an upper or lower limit.

Figure 4. I found the label "joint-probability histogram" for the color bar confusing. Is this the relative/normalized number density of particles observed in each bin? Please clarify what you mean here.

p. 12 line 371. Is this number fraction of evaporating tar particles fairly consistent over the observed optical size range for the tar-rich fire? What was the optical size

distribution of evaporating tar particles observable by the SP2? Figure 4 suggests that these particles were in general larger than rBC; is this only true for the tar BrC associated with incandescence?

p. 13 line 402-403. This percentage could depend on the SP2 laser power and may not be illustrative of all operational conditions. Can you also provide a percentage for the portion of non-incandescing particles observed by the SP2 that were identified as tar BrC?

---

## Author Comment (AC1) · 24 Oct 2019

**Reviewer responses by Corbin and Gysel-Beer for *Detection of tar brown carbon with the single particle soot photometer* doi:10.5194/acp-2019-568**

**Response to Reviewer #1**

*The current manuscript builds upon recent findings that the SP2 can induce some NIR light absorbing organic*

*aerosol particles (commonly referred to as brown carbon or BrC) to char and subsequently be detected as refractory black carbon in the SP2. Specifically, the present manuscript examines tar brown carbon particles (referred to as "tar brC") that are produced by a marine engine running on heavy fuel oil. Using both the time-dependent scattering and incandescence channels of the SP2, the authors report that tar BrC annealed by the SP2 laser, and then detected as "rBC", can be distinguished from primary emission rBC from incomplete combustion. This new*

*analysis methodology provides a framework that will expand the utility of the SP2 to include the subclass of BrC particles that absorb at the lasing wavelength of the SP2 (1064 nm). The methodology and analysis presented are sound and the paper is written well. Publication is certainly warranted, once the authors address a concern on material definition and clarify a nomenclature issue as highlighted below.*

Reviewer #1 has given an excellent review, pointing out places for improvement in our literature review and for consistency in our discussion. We are grateful to the Reviewer for their time and valued input. We have revised the Review section of our manuscript extensively following this input.

*With respect to material definition, a core statement that the authors make, and one that traces back to the lead author's 2019 Climate and Atmospheric Science paper, is that tar ball material is insoluble (e.g., cited on page 2, line 27 and again page 3, line 89- 90 in present manuscript). The authors are encouraged to look take a second look*

*at the closing section in Hand et al., 2005 where these authors state (page 12; "..ESEM experiments of particle hygroscopicity indicate that tar balls observed during YACS are water soluble at high relative humidity (RH > 83%), albeit to a much lower degree than inorganic salt particles and with no distinct deliquescence point."*

Indeed, our assertion of insolubility traces back to our 2019 Clim Atmos Sci paper. The present manuscript arose out of our efforts, while preparing that paper, to use the SP2 data more quantitatively in analyzing tar particles.

The reviewer's criticism here led us to carefully reconsider the meaning and significance of the insolubility of tarballs. We have revised the entire Review section as a result. Our revisions emphasize that tarballs form via continuous processes, so that a given sample may fall on any point of this continuum. Naturally, intermediate materials will be observed. A line must be drawn somewhere along this continuum in order to define tarballs meaningfully. Even in the case of the original electron-beam stability definition of Posfai et al., some time scale must be specified for stability, and an intermediate stage would exist where particles evaporate/sublimate slowly rather than negligibly in the beam.

Due to the inherent connection between the degree of polymerization and/or carbonization and solubility, we stand by our recommendation that solubility be viewed as a useful dividing line to specify that a particle is a tarball, in the event that its stability in an electron microscope has not been measured. The value of this distinction is that traditional methods for measuring brC (solvent extraction) are insensitive to tar brC; and that traditional methods for measuring BC may be sensitive to tar brC

(Corbin et al., 2019). Failing to recognize these points may result in confusion and misunderstanding of the environmental (climate and health) impacts of tar brC. We acknowledge that a continuum of materials may exist, and emphasize that great care must be taken by future studies to take this fact into account, but drawing the line at solubility has important implications for climate (it will cause the accumulation of tarballs on the surface of melting snow) and health (insoluble PM will not dissolve in the lungs and can only be cleared by macrophages). It also has implications for light absorption: our definition is biased towards the more-carbonized and therefore more-light-absorbing side of the continuum, and light absorption is a major reason why tarballs are the focus of much recent research.

In our updated Section 2 we now note that the evidence of TB hygroscopicity presented by Hand et al. (2005) is consistent with the view that TBs of a wide range of maturities are present in the atmosphere. Hand et al. (2005) do indeed demonstrate TB hygroscopicity for relatively aged particles. The degree of aging and light absorption in their samples is unknown. Based on our review and the work of Hoffer et al. (2016), we infer that their particles would be less light absorbing. We thank the reviewer for emphasizing this work.

The existence of soluble tarball-like particles does not actually contradict our main message. The most graphitic tarballs will be the most absorbing, so if a dividing line between brown carbon droplets and TBs is to be drawn, then it may be appropriate to draw this line based on solubility. (Though we use the term graphitic here we emphasize that tarballs are not capable of becoming fully graphitized nor as graphitized as soot.) At the same time, allowing for this important caveat to be recognized is an important point which the reviewer has made well, and which our submitted manuscript did not adequately acknowledge. This reviewer comment is closely related to the next comment regarding Li et al. (2019) and we respond in much more detail there.

*Additionally, this reviewer wishes to draw attention to the recent work of Li et al., (ACP, 19, 139-163, 2019) where*

*the presence of both polar and non-polar TBs (e.g., water soluble and non-soluble components) are reported on and discussed. The observation that wildfire TBs are water soluble and that some laboratory-generated TBs can be as well, suggest that care should be exercised with the application of generalized labels such as that used in the present manuscript. While the tar brC particles investigated in the present study may very well be completely insoluble, the reported presence of water-soluble TBs suggests that there are in indeed two classes of "tar-like" particles and*

*further suggests that some reconciling is in order.*

The recent work of Li et al. (doi:10.5194/acp-19-139-2019) is an example of an excellent laboratory study which provides fundamentally valuable information about the refractive index of laboratory-generated "tarballs". Our failure to cite it was an accident of oversight as Li et al. was published after the bulk of our manuscript was written. Regardless, we do not find that this paper represents the same "tarballs" studied by previous work or first identified by Posfai et al. (2004).

Li et al. provided fundamental and valuable information on wood-pyrolysis organics (which in some communities might be called "tar" in the sense of a dark and highly viscous liquid) but they did not include a heat-processing stage (called "heat shock" by Tóth and coworkers) in their TB preparation protocol and therefore studied a less-graphitized material than has been observed in some studies such as Alexander et al. (2008) and Hoffer et al. (2016a). That is, Li et al. could be said to have studied a material closer to "brown carbon" than TBs, to the extent that such a distinction is meaningful (the lack of a clear distinction is now mentioned in the revised Section 2), and this is supported by the fact that their reported refractive indices are similar to those of soluble brC (see summary table in their paper). As argued in the present manuscript (and in Corbin et al. npj Clim Atmos Sci 2019, doi:10.1038/s41612-019-0069-5), the literature does suggest that such a distinction is meaningful and useful in interpreting both measurements and environmental fates, even if the dividing lines between brC and TBs are fuzzy and occur over multiple physico-chemical dimensions, as noted above.

The heat-processing stage mentioned above was described in the original Tóth et al. (2014) methodology otherwise followed by Li et al (2019). The polar-and non-polar "TBs" which Li et al. studied were generated by nebulizing the two phases of the emulsion produced by by heating wood pellets to $530\,°C$ at $25\,°C\,min^{-1}$ and then holding for 20 minutes. Li et al. (2019) deviated from the methodology of Toth et al. (2014) and Hoffer et al. (2016) by omitting the heat-shock stage of the Toth et al. method. That is, Toth et al. and Hoffer et al. passed the generated particles through an oven set to $600\,°C$ with a 0.3 second residence time. Toth et al. showed using electron microscopy that without heat shock, the generated particles were liquid-like rather than solid, according to the observation that liquid-like particles are able to wet the electron microscopy substrate and thus deform from an initially spherical morphology. Li et al. (2019) presented electron microscope images showing that even without heat shock, their particles appeared spherical in the electron microscope, which suggests that their TBs were of a different composition to those of Toth et al., presumably due to a difference in the starting material (Oberlin, 1984).

Hoffer et al. showed that the heat-shock procedure resulted in lowering the Ångström absorption exponent (AAE, calculated for wavelengths between 467 and 652 nm) of their generated particles from about 5 (a typical value for soluble brown carbon) to about 3 (much lower than soluble brC and consistent with field results on TBs). Moreover, a heat-shock temperature of well over $500\,°C$ was needed to achieve this change in AAE. Therefore, a fundamental change in the material properties of the "tar" occurred upon heat shock, such that the two studies will have addressed materials with different light absorption properties. Yet,

Li et al. (2019) do present electron microscopy images showing spherical particles, whereas Tóth et al. observed non-spherical particles in their electron microscope (due to the spreading of particles on the substrate, indicating a low viscosity of those particles) which further indicates the variability in TB composition which may result from different starting materials.

The protocol of Li et al. also differed slightly from the original method by adding an evaporation stage by heating to $300\,°C$ under nitrogen. Since this temperature is well below $500\,°C$ it would be unlikely to have had any effect on the results.

We have gone into depth on the comparison of Li et al. (2019) with TB literature because we think it is an excellent example of a very important point: *when studying complex carbonaceous materials on the continuum of carbonization (or of volatility, solubility, or light-absorption), studies which have the same starting material and apply the same measurement techniques but slightly different processing conditions may not actually be looking at the comparable points on those continua.*

In other words, while we acknowledge the possibility that the TBs produced from residual fuels are different to the TBs produced by biomass, we emphasize that it has also been demonstrated that even biomass-TBs produced by one process may differ hugely from biomass-TBs produced by another procedure. We agree with the reviewer's reservations that care is needed, and we have deeply considered our choices of terminology before deciding that the terminology that is most likely to minimize future confusion is to refer to our residual-fuel, tarball-like particles as tar brC.

We are prepared to adjust our terminology as further evidence arises in future studies, but, in an important sense, more care is needed for biomass-TB studies than for our study, since any reader will notice the difference in fuels for our work, whereas few readers will notice important but subtle changes in experimental protocols for biomass-tarball studies, which can have major impacts on properties such as the MAC and AAE. Therefore, although we have not used the same starting material as the TB literature, we have argued from fundamental physico-chemical principles for categorizing these materials in the same group.

**Recent work by Adler et al. (2019)**

Reviewer #1 did not mention this recent work but it was published during the review of our paper, and is relevant to the discussion here. Adler et al. (2019, doi:10.1080/02786826.2019.1617832) studied both laboratory biomass-burning smoke emissions and wildfire smoke. They identified spherical, refractory particles of amorphous carbon, stable under the electron beam of an electron microscope, which did not incandesce in an SP2.

The discussion of Adler et al. (2019) carefully avoids the use of the term TBs, presumably because the authors of that work wished to limit the definition of TBs to the alternatives discussed in our manuscript, rather than to the materials-based definition we are proposing. Whereas Adler et al. (2019) concluded that more work is needed to understand whether their particles correspond to TBs, we carefully considered their evidence and found that they have actually evaluated *all* of the properties used to define tarballs. Only the atmospheric aging definition is not satisfied by their samples. In other words, we do not find that Adler et al. (2019) could *possibly* have measured tarballs; we find that Adler et al. (2019) have *by definition* measured tarballs.

Importantly, Adler et al. (2019) also found that in their samples, where tarball absorption represented a substantial fraction of total light absorption, tarballs were insoluble (that is, measurements of water-soluble brown carbon could not explain the difference between total absorption and BC-related absorption, so tarballs were insoluble).

In summary, the work of Adler et al. (2019) strongly supports our recommendations because they demonstrate that strongly light-absorbing tarballs in wildfires were insoluble, present in the smoke prior to aging. By alternative non-materials-based definitions of TBs, the particles described by Adler et al. would require a new nomenclature for a material which is chemically and physically indistinguishable from TBs, and which would not reduce ambiguity any more than the terms "wildfire tarballs" or "residual-fuel tarballs" (or, when of unidentified origin, simply tarballs) would provide.

*The fuel source for a marine engine is very different from that available for wildfires - not to mention the emission sensitivity to burn conditions in the latter - so perhaps we should not be too surprised that there could be some fundamental differences of specific particle types within this subclass of brown carbon particles. The authors need to address this and adjust their material definition argument accordingly. If indeed biomass burning tar particles are different from fossil fuel tar particles, how generalizable are the conclusions about tar brC behavior in the SP2 laser?*

We have hypothesized that TBs from biomass and residual fuels would be similar, since residual fuel (fuel containing the heaviest fraction of crude oil) actually originates from biomass. That is, crude oil is biomass subjected to slow, pressure- and temperature-related chemical transformations. These transformations will obviously not be identical to the rapid, temperature-related chemical transformations of combustion, yet may be related. For example, if both transformations proceed via kinetically-limited carbonization, as proposed by Corbin et al. (2019), then it is the molecular structure of the starting material that dictates the reaction endpoint, and not the precise conditions of the reaction. This is shown by Oberlin (1984), as now mentioned in the text (around line 145 of the revised manuscript).

A separate reason why residual-fuel TBs and wildfire TBs may be best placed into the same category is that the material referred to as "TBs" spans a wide range of properties. The wider this range, the more likely it is to capture the TBs produced by these two different sources. We base this conclusion on the present literature on wildfire TBs, which span a wide range as discussed above, and acknowledge that we may revise this conclusion as that literature begins to better constrain the likely properties of wildfire TBs.

We note also that the laser power used by the SP2 affects whether or not particles are observed to incandesce by that instrument. We believe that the reviewer had this in mind when writing his or her comment. This power will need to be standardized between studies, if the approaches used in recent SP2 work are to be compared.

We have referred to line 145 above for the corresponding revisions to our text. To avoid confusion, we have not copied the revised text here – the entire section has been revised.

*This reviewer is confused by the presence of a non-incandescing distribution for the "soot BC case" (blue line) in Figure 2. In section 3.2, the authors reference a "pure-BC case" as a "control" to help identify the unique features of the tar particles, yet in the figure, they cite "soot BC case". It is not clear whether the authors are referencing two different "BC" controls or are using two labels for the same material. My concern is that a "pure-BC case" should be composed of solely rBC particles that would all incandesce in the SP2 and that would necessarily lead to a non-Gaussian scattering signal distribution for in Figure 2. Additionally, the traces presented in Figure 2 are presumably for non-incandescing particles, which, again, for a pure-BC case, no pure scattering signals should be present. In short, clarification is in order. I believe the solution is straightforward enough. As this reviewer understands the manuscript text, the "soot BC case" are those particles generated when the marine engine is running under a specific operation condition and hence serves as a control on the particle-type produced and believes that their reference to "pure-BC case" is the same material. If correct, please clarify this nomenclature and stick to one label for your control material.*

The Reviewer is correct here, we were not precise enough in our terminology.

The reviewer's interpretation is correct ("as this reviewer understands...") and we edited the text accordingly.

All instances of "soot case", "tar case", or similar have been changed to one of "tar-free case" or "tar-rich case", to avoid implying that only soot or tar were present in either case.

We also added arrows and a label to the figure to explain what the non-incandescing distribution is. In the submitted version, the label was present, but may have been misinterpreted by a reader as applying only to the tar-rich case.

*Other specific issues: Page 1, lines 5 -6. The authors write ": : :.tar brC results in unique SP2 signals due to a combination of complete or partial evaporation, with no or very little incandescence. Approximately 70% of tar particles incandesced." The juxtapositioning of these two sentences is very awkward. The first seems to say that there is very little to no incandescence while the follow on sentence says that 70% of the particles incandesced. Which is it? Please reword.*

We clarified by changing this text to: [...] Only a subset of tar brC particles exhibited detectable incandescence (70% by number); for these particles the ratio of incandescence to light scattering was much lower than that of soot BC.

*Page 4, line 95, 96. The authors are encourage to examine (and possibly reference, if they think it is relevant) the recent paper by Li et al., (ACP, 19, 139-163 (2019)) where the optical properties of laboratory-generated TBs were investigated and refractive indices derived.*

We regret omitting to cite this paper by Li et al. and are very grateful to the reviewer for pointing it out. (As evidenced by our extremely long discussion of this paper above.) In fact, this paper led us to substantially modify our discussion of the material definition of tarballs, because its implicit definition of tar balls (nanoparticles produced by the nebulization of the viscous liquid produced by dry distillation of crushed wood pellets) is fundamentally different to that of Posfai et al. (spherical carbonaceous particles resistant to electron beam damage).

This comment has also led us to overall modify the review of tar brC properties in Section 2.1. We caution here that the particles studied by Li et al. (2019) should be considered as on the lower end of possible absorptivities for laboratory-generated TBs.

*Page 6, line 183-185. The $\sim 3000\,K$ refractory requirement is dictated by the bandpass filters used in the commercial-grade SP2. A different set of bandpass filters would enable a lower temperature blackbody to be detected. (Admittedly, this reviewer is being a bit pedantic here, but still...)*

We disagree somewhat, as the detection of incandescence relies not only on the blackbody temperature but also on the absolute radiance of the heated particle, which decreases with decreasing temperature. So, the instrument may also require enhanced sensitivity to detect a substantially lower blackbody temperature. The spectral response of the detector (photomultiplier tube) would also come into play. Since the paragraph began with the sentence **At least two conditions must be met for particles to incandesce in the SP2.** we consider that we have already constrained the discussion to the overall instrument performance, and cannot see how the paragraph could be made clearer overall. We appreciate the reviewer's attention to detail here!

*Page 18, Figure 4. While the authors discuss how different thickly-coated rBC particles would look cast on a similar plot as shown for two cases in Figure 4, this reviewer wonders whether it would be useful to have a third panel on*

*this figure for thickly-coated particles. As a picture is worth a thousand words, so too is a plot worth a thousand arguments and thus might prove a useful visual comparison for those readers who are not well acquainted with the SP2 mixing state analysis and data products. This is certainly not critical, but rather a suggestion, and thus at the authors discretion.*

This issue is not as complex as it seems, but is rather the result of our aiming to use accurate language. An explanation follows below. We have reworded the Figure caption to be more understandable and do appreciate the Reviewer's pointing out this confusion.

The short answer to this ostensibly complex issue is that Figure 4 plots the ratio of particle size measured by scattering to particle size measured by incandescence, at the time of incandescence. Given that we plotted a ratio of soot-equivalent diameters, the naïve prediction for uncoated soot particles in Figure 4 is the line $y = 1$. But thickly coated soot particles also fall on the line $y = 1$, because the coating influences the initial optical diameter but not the optical diameter at incandescence. For extremely-thickly coated soot particles, particle breakup may prevent total evaporation of coatings, which would cause additional scattering and move the particles above the line $y = 1$. This is a potential cause of confusion when trying to distinguish extremely-thickly coated soot from tar. The difference that would be observed between these two species is that tar particles evaporate after incandescence, whereas a "coating fragment" (droplet resulting from breakup of a thickly-coated soot particle) would not evaporate after incandescence. In terms of scattering cross-section $C(t)$, the coating fragment would give a constant signal whereas the tar particle would give a decreasing signal. These kind of details are the reason we recommended a machine-learning approach to tar detection like that of Lamb (2018) in our conclusions.

A more technical response is that we plotted the ratio of the quantities which the PSI SP2 Toolkit calls "scattering at incandescence" and "BC mass", after converting both quantities to equivalent *soot BC* diameters according to the best constraints available to us (described in the manuscript and also in Corbin et al., 2018b). From this perspective, one may predict the trends in terms of $y$ described above.

This information was briefly conveyed in the manuscript in the paragraph ending with:

> For clarity we have therefore applied these calibrations to the data presented in Figure 4, although they do not apply to tar.

There are two reasons why we avoided discussing this in detail in the manuscript, although we do allude to it briefly. First, the conversion to diameter requires assumptions of refractive index and mass-specific incandescence ("incandescence calibration"). Since these properties are poorly constrained for tar, and cannot be determined prior to deciding whether or not a particle is a tar particle, we avoided this strategy and labelled the ordinate units on Figure 4 as arbitrary. It also gets confusing when trying to decide where tar should move on this plot (it should scatter more, but it may have different optical properties. It would be possible to model this using reported refractive indices for tar, but the complexity of such a calculation would exceed its scientific value.)

Second, in reality, for large soot particles (with optical size parameter $\gg 1$), the Rayleigh-Debye-Gans approximation breaks down and particles begin to scatter less light than predicted from their volume, so the abovementioned ratio of diameters deviates downwards from 1. (Note that we have not specified a definition of "size" in the previous sentence; for open-structured soot aggregates the optical size parameter may relate more closely to their monomer diameter than to their volume-equivalent diameter, though this relationship eventually breaks down.) This is visible in our Figure 4, where the soot-BC ratio deviates downwards. it was also shown systematically by Moteki and Kondo (2010, doi:10.1080/02786826.2010.484450; Figure 9a).

Rather than add all of this discussion to our paper we have tried to rephrase the legend of Figure 4 to convey the same message. We have not changed the actual figure; we considered drawing a line at $y = 1$ which later curves downwards, but we have no constraints on the position or rate of curvature and wish to avoid misleading readers.

We changed the legend to:

*Incandescing tar particles distinguished from soot BC by their scattering properties in the SP2. The abscissa shows the optical particle diameter prior to evaporation, calculated from $C(t_i)$ with $m = (1.5, 0)$. The ordinate shows the ratio of optical-equivalent and incandescent-equivalent particle diameters, $C(t_i)/I(t_i)$, calculated with the optical properties of soot BC. The colour scale shows the number fraction of particles in each bin (that is, a normalized joint probability histogram). Note that $C(t_i)/I(t_i)$ employs the optical diameter at time of incandescence $t_i$ and therefore represents the optical diameter of the refractory component of the particle. Therefore, for soot BC $C(t_i)/I(t_i) \approx 1$ by definition, although deviations below unity are observed for larger particles where light-scattering transitions from from the Rayleigh regime to the geometric regime (Moteki and Kondo, 2010). Thickly-coated soot BC particles are also expected to fall on the $C(t_i)/I(t_i) \approx 1$ line since volatile coatings evaporate before $C(t_i)$ is measured, as discussed further in the text. In contrast, $C(t_i)/I(t_i)$ is very different from unity in the presence of particles which contain non-incandescing but refractory material.*

...from the old version:

*Scattering cross-section just prior to incandescence ($C(t_i)$; see Figure 1 for illustration) normalized to maximum incandescence signal $I(t_i)$, plotted as a function of particle optical diameter prior to evaporation. Under the condition that the incandescing material is comparable in all cases, an increase in $C(t_i)/I(t_i)$ indicates the presence of non-BC refractory material. Inspection of the data indicated that the joint probability is not exactly equal to zero in the soot-BC case due to coincidence (simultaneous presence of two particles in the laser beam).*

The "further discussion" referenced in the text was previously a single sentence and has been expanded on:

*While soot coatings ideally undergo complete evaporation in the SP2 prior to incandescence, extremely-thick coatings may result in particle breakup (Moteki and Kondo, 2007; Sedlacek et al., 2012; Dahlkötter et al., 2014). That is, the coating may fragment and generate a secondary particle large enough to generate a scattering signal in the SP2. This fragment particle would not be in thermal contact with the rBC core and would therefore be characterized by the example shown in Figure 1a. The fragment particle would cause additional scattering at the time of incandescence, and shift the resulting signal towards higher $C(t_o)/I_{peak}$ values in Figure 4. Moteki and Kondo (2007) observed that graphite particles coated with oleic acid or glycerol did not undergo fragmentation until*

*initial diameters of 400 nm or 600 nm, respectively. The fraction of fragmenting particles then increased rapidly until virtually all particles fragmented at 500 nm or 650 nm, respectively. Since no such rapid transition is seen in Figure 4, and since our data set employed a tar-free control case, we can be confident that fragmentation did not play a role in our data set. Moreover, as mentioned earlier, we have verified by manual inspection of our data that the particles labelled "tar" in Figure 4 were not thickly-coated and that no scattering signal remained after incandescence.*

We also noticed an error in our Figure 4 legend, $C(t_i)$ was erroneously labelled as $C(t_o)$.

**Response to Reviewer #2**

*Summary: The authors present evidence that a material they identify as tar brown carbon (tar BrC) has a unique signal in the SP2 (evaporation but little/no incandescence) based on observations of emissions from ship engines. They use the time dependent scattering channels and broad-band incandescent channel in the SP2 to provide a method to identify these tar BrC particles. These observations provide additional evidence of further utility of the SP2 in identifying other aerosol types than rBC, and provides a new on-line measurement technique for a specific class of brown carbon that absorbs at the wavelength of the SP2 laser. The analysis approach for identifying the tar BrC is clearly described, and the authors also thoroughly investigate an alternative explanation for the tar BrC associated with low incandescence (if it might instead be thickly coated rBC). The writing is very clear and the manuscript is logically written. This paper will be a valuable addition to the literature once a few minor points have been addressed.*

We thank the reviewer for their detailed review and constructive feedback, which has allowed us to improve the quality of our manuscript. Thank you for your time.

*General Comments: I would like to see more description of the data sets and observations used in the study; although this paper referenced Corbin et al. 2019 (where these measurements were described in more detail), it would make this work more useful as a stand-alone paper to have some additional details of the observations given here. These details would also help the reader better understand how these specific observations of aerosols from ship engine emissions might relate to other aerosol sources, such as biomass burning. Additionally, the authors have pointed out that the in-SP2 annealing that has been observed in Sedlacek et al. 2018 but not in Moteki and Kondo 2008 may be due to differences in experimental procedures; thus, a greater description of the SP2-specific calibrations in this work would be warranted.*

These general comments are all fair and have led to improvements to our manuscript. In some cases we responded slightly differently, for example, we aim to help the reader better understand how ship emissions might relate to biomass burning by expanding our Discussion section. We made the following changes:

In response to the request for more description of the data sets, we have doubled the length of Section 3.2, where our test data set was described.

We subdivided the discussion with subsections and expanded on some of the topics, including a subsection on laser annealing of tar in the SP2, detection of tar brC by other real-time instruments, relative number of incandescing and non-incandescing tar particles (where the issue of SP2-specific calibrations/configurations is addressed), and comparison with biomass-burning tarballs.

We did provide a detailed technical description of the SP2 but we agree that our comment of the difference between Sedlacek's and Moteki's observations was too vague. We have revised it from This may reflect variability in the experimental procedures or in the composition of the nigrosin. to **This may be due to the use of a lower laser intensity by Moteki et al., which was** **shown by Sedlacek et al. to potentially result in negligible incandescence.**

*I share some of the concerns of Referee #1, that specifically referring to these aerosols detected by the SP2 as "tar BrC" may be mis-leading, given the significant literature surrounding tar balls, and the fact that the aerosols in this current study are identified solely based on their observed signals in the SP2. However, as the authors have taken the time to clarify this in Section 2, I do not have major concerns about this terminology.*

The particles discussed in this study were not identified solely on their SP2 signals!

They were identified in Corbin et al. (Clim Atmos Sci 2019) as being insoluble, spherical, having a wavelength dependence of absorption greater than unity that would render them brown. Moreover, they were identified by electron microscopy as stable under the electron beam. We have clarified the text by adding

Tarballs were verified as present in this data set using electron microscopy (Corbin et al., 2019), and were quantified by a combination of techniques. Since those techniques were not specific to spherical particles, and since tarballs may have existed as agglomerates with other material, we have used the term "tar brC" to refer to the material (as discussed in Section 2).

Our particles possessed all of the known and expected properties of highly-absorbing tar balls without exception. It is therefore very unlikely that these particles are composed of a material substantially different to biomass-burning TBs. Given that these two materials are virtually the same, they should be referred to using the same name to avoid confusion. A qualifier such as "residual-fuel tarballs" versus "wildfire tarballs" may be appropriate when referring to both sources.

At the end of Section 2, we have added the following sentence as well as various minor changes to account for the good point made here by the Reviewer:

**The broader category of tar brC may include subcategories for biomass- and residual-fuel-related particles,** **if future work shows that this is necessary, as well as minor categories to account for details such as the** **formation of oxygenated interfaces due to atmospheric processing (Tivanski et al., 2007).**

Similarly we added the **boldface** sentence of the following text to pre-existing text in Discussion, and added a Discussion sub-heading "Comparability with tarballs described by other studies" to emphasize this issue:

Care must therefore be taken when extrapolating the present results to other studies. Certain tar samples may be less carbonized (and less likely to undergo laser-induced evaporation) or more carbonized (and more likely to undergo laser-induced incandescence) than our samples. **This includes residual-fuel tar brC samples produced** **by different engines or different combustion systems, as well as biomass-burning tar brC.** Based on the wavelength-dependence of absorption of tar in our samples reported and placed in the context of literature by Corbin et al. (2019) (AAE of about 3), we believe that our tar samples were of a typical degree of carbonization.

**Moreover, in all studies on light-absorbing carbonaceous particles, it is essential to report not only a name for the particles being studied, but as many properties as is practical, and as are justified by the novelty of the particle source. These properties include light absorption efficiency (MAC), wavelength dependence (AAE), morphology (which directly influences light absorption), volatility and solubility in relevant solvents.**

*Specific Comments: p.7 Section 3.2. Please clarify whether the SP2 operating conditions were the same during observations of both the pure-BC case and the tar-rich case, as this would impact the interpretation of the experiments.*

We added **with no changes to the SP2.** to the sentence **The two cases were measured on the same day using the same sampling configuration with no changes to the SP2.**

*p.8 line 250-252. Clarify if 160 nm is an upper or lower limit.*

We added "lower limit". This lower size limit of optical sizing is a simple issue of signal-to-noise at the detector.

     *Figure 4. I found the label "joint-probability histogram" for the color bar confusing. Is this the relative/normalized number density of particles observed in each bin? Please clarify what you mean here.*

The reviewer is correct, joint-probability histogram is the technical term for a two-variable probability distribution (and the
name for the Igor Pro function which we used to generate the plot). We changed the label to the equally correct "Number fraction" and specify that the data are a joint probability histogram in the text.

     *p. 12 line 371. Is this number fraction of evaporating tar particles fairly consistent over the observed optical size range for the tar-rich fire? What was the optical size distribution of evaporating tar particles observable by the SP2? Figure 4 suggests that these particles were in general larger than rBC; is this only true for the tar BrC associated*
*with incandescence?*

We did not perform a size-resolved analysis of tar particles due to limitations in our data. The main limitation is that we could only confidently distinguish between evaporating and non-evaporating particles for particles larger than 180 nm. This value is only slightly larger than the lower limit of quantification of the SP2 for optical sizing *at peak scattering intensity*. It is larger due to the need to reliably measure scattering intensity at less-than-peak intensities, since that was the goal of our analysis.
In addition to this sensitivity-limited lower bound, we were limited at the upper size range by a lack of large particles in our data set. We therefore specified an upper bound to clarify to the reader that 400 or 500 nm particles were not represented by our analysis.

This size range was labelled on the ordinate axis of Figure 2, but we have now also written it in the legend.

     *p. 13 line 402-403. This percentage could depend on the SP2 laser power and may not be illustrative of all*
*operational conditions. Can you also provide a percentage for the portion of non-incandescing particles observed by the SP2 that were identified as tar BrC?*

The reviewer's first point is an important point which our addition of subheadings in the Discussion section should help to emphasize (in particular, Section 5.1).

The reviewer's second point is unfortunately not possible in our data set due to the presence of large amounts of lubrication-oil related particles. These particles are known to be volatile but are produced by a completely independent mechanism than tar (which is produced by the ejection of low-volatility material from HFO droplets), so the relative number of these two species is meaningless. In a future experiment where the SP2 would be operated after a thermal denuder, it may be possible to obtain a meaningful number. Unfortunately, we did not do this experiment.

We added the following text to the conclusions:

**The number fraction of evaporating, non-incandescing particles (evaporating tar) relative to all tar particles or simply relative to all particles is not reported due to the presence of significant amounts of lubrication-oil related particles (Eichler et al., 2017), which would bias this number fraction low by an unconstrained amount.**

**Other changes to the manuscript**

We have added graphics to Figure 1 depicting the morphology and nature of the particles represented by each sub-panel.